# Novel cyclic homogeneous oscillation detection method for high accuracy and specific characterization of neural dynamics

**Hohyun Cho[1,2]\*, Markus Adamek[1,2], Jon T Willie[1,2], Peter Brunner[1,2]\***

[1]Department of Neurosurgery, Washington University School of Medicine, St. Louis, United States; [2]National Center for Adaptive Neurotechnologies, St. Louis, United States

**\*For correspondence:**
hohyun@wustl.edu (HC);
pbrunner@wustl.edu (PB)

**Abstract** Determining the presence and frequency of neural oscillations is essential to understanding dynamic brain function. Traditional methods that detect peaks over $1/f$ noise within the power spectrum fail to distinguish between the fundamental frequency and harmonics of often highly non-sinusoidal neural oscillations. To overcome this limitation, we define fundamental criteria that characterize neural oscillations and introduce the cyclic homogeneous oscillation (CHO) detection method. We implemented these criteria based on an autocorrelation approach to determine an oscillation's fundamental frequency. We evaluated CHO by verifying its performance on simulated non-sinusoidal oscillatory bursts and validated its ability to determine the fundamental frequency of neural oscillations in electrocorticographic (ECoG), electroencephalographic (EEG), and stereoelectroencephalographic (SEEG) signals recorded from 27 human subjects. Our results demonstrate that CHO outperforms conventional techniques in accurately detecting oscillations. In summary, CHO demonstrates high precision and specificity in detecting neural oscillations in time and frequency domains. The method's specificity enables the detailed study of non-sinusoidal characteristics of oscillations, such as the degree of asymmetry and waveform of an oscillation. Furthermore, CHO can be applied to identify how neural oscillations govern interactions throughout the brain and to determine oscillatory biomarkers that index abnormal brain function.

## eLife assessment

Building on previous toolboxes to distinguish 1/f noise from oscillatory activity, this study introduces an **important** advancement in neural signal analysis to identify oscillatory activity in electrophysiological data that refines the accuracy of identifying non-sinusoidal neural oscillations. Extensive validation, using synthetic and various empirical data, provides **convincing** evidence for the accuracy of the method and outlines practical implications for relevant scientific problems in the field.

## Introduction

Neural oscillations in the mammalian brain are thought to play an important role in coordinating neural activity across different brain regions, allowing for the integration of sensory information, the control of motor movements, and the maintenance of cognitive functions (*Pfurtscheller and Lopes da Silva, 1999*; *Caplan et al., 2003*; *Buzsáki and Draguhn, 2004*; *Jensen and Mazaheri, 2010*; *Giraud and Poeppel, 2012*; *Schalk, 2015*; *Fries, 2015*). Detecting neural oscillations is important in neuroscience as it helps unravel the mysteries of brain function, understand brain disorders, investigate cognitive

processes, track neurodevelopment, develop brain-computer interfaces, and explore new therapeutic approaches. Thus, detecting and analyzing the 'when', the 'where', and the 'what' of neural oscillations is an essential step in understanding the processes that govern neural oscillations.

For example, detecting the onset and offset of a neural oscillation (i.e. the 'when') is necessary to understand the relationship between oscillatory power/phase and neural excitation, an essential step in explaining an oscillation's excitatory or inhibitory function (*Pfurtscheller and Lopes da Silva, 1999*; *Canolty et al., 2006*; *Klimesch et al., 2007*; *Haegens et al., 2011*; *de Pesters et al., 2016*). Localizing the brain area or layer that generates the oscillation (i.e. the 'where') provides neuroanatomical relevance to cognitive and behavioral functions (*Buzsáki and Draguhn, 2004*; *Miller et al., 2010*). Lastly, determining the oscillation's fundamental frequency (i.e. the 'what') indicates underlying brain states (*Penfield and Jasper, 1954*; *Buzsáki and Draguhn, 2004*). Together, the 'when', the 'where', and the 'what' can be seen as the fundamental pillars in investigating the role of oscillations in interregional communication throughout the brain (*Fries, 2015*). These fundamental pillars can also provide insight into the functional purpose (i.e. the 'why'), underlying mechanisms (i.e. the 'how'), and pathologies (i.e. the 'whom') of neural oscillations (*Buzsáki and Draguhn, 2004*; *Buzsaki, 2006*).

The detection of neural oscillations has historically been extensively studied in the frequency (*Wen and Liu, 2016*; *Donoghue et al., 2020*; *Ostlund et al., 2022*), time (*Hughes et al., 2012*; *Gips et al., 2017*), and time-frequency domains (*Chen et al., 2013*; *Wilson et al., 2022*; *Neymotin et al., 2022*). With the notable exception of *Gips et al., 2017*, these studies assume that neural oscillations are predominantly sinusoidal and stationary in their frequency. However, there is an increasing realization that neural oscillations are actually non-sinusoidal and exhibit spurious phase-amplitude coupling (*Belluscio et al., 2012*; *Cole et al., 2017*; *Scheffer-Teixeira and Tort, 2016*; *Gips et al., 2017*; *Donoghue et al., 2022*). A recent review paper on methodological issues in analyzing neural oscillations (*Donoghue et al., 2022*) identified the fundamental frequency of non-sinusoidal neural oscillations as *the most challenging problem* in building an understanding of how neural oscillations govern interactions throughout the brain.

Fast Fourier transform (FFT) is the most commonly used method to detect neural oscillations. The FFT separates a neural signal into sinusoidal components within canonical bands of the frequency spectrum (e.g. theta, alpha, beta). The components of these canonical bands are typically considered to be functionally independent and involved in different brain functions. However, when applied to non-sinusoidal neural signals, the FFT produces harmonic phase-locked components at multiples of the fundamental frequency. While the asymmetric nature of the fundamental oscillation can be of great physiological relevance (*Mazaheri and Jensen, 2008*; *Cole et al., 2017*; *Donoghue et al., 2022*), its harmonics are considered to be an artifact produced by the FFT that can confound the detection and physiological interpretation of neural oscillation (*Belluscio et al., 2012*; *Donoghue et al., 2022*).

An example of an unfiltered electrocorticographic (ECoG) recording from auditory cortex (*Figure 1A*) illustrates the non-sinusoidal nature of neural oscillations. The associated FFT-based power spectrum (*Figure 1B*) exhibits multiple peaks over $1/f$ noise even though only one oscillatory signal is visibly present in the time domain signal. Whether the peaks over $1/f$ at 12 and 18Hz, are harmonics of 6Hz oscillations or independent oscillations remains unknown. This ambiguity affects the ability to accurately and efficiently identify neural oscillations and understand their role in cognition and behavior. For this illustrative example of non-sinusoidal neural oscillations, we used a phase-phase coupling analysis (*Belluscio et al., 2012*) to determine whether the exhibited 18Hz beta oscillation is a harmonic of the 6Hz theta oscillation. This analysis confirmed that the beta oscillation was indeed a harmonic of the theta oscillation (*Figure 1E and F*). In marked contrast, for a sinusoidal neural oscillation, a phase-phase coupling analysis could not fully ascertain whether the oscillations are phase-locked and thus are harmonics of each other (*Figure 1G–L*). This ambiguity combined with the exorbitant computational complexity of the entailed permutation test and the requirement to perform the analysis across all cross-frequency bands over all channels and trials render phase-phase coupling impracticable for determining the fundamental frequency of neural oscillations in real time and, thus, the use in closed-loop neuromodulation applications.

In this study, we aim to define the principle criteria that characterize a neural oscillation and to synthesize these criteria into a method that accurately determines the duration ('when'), location ('where'), and fundamental frequency ('what') of non-sinusoidal neural oscillations. For this purpose,

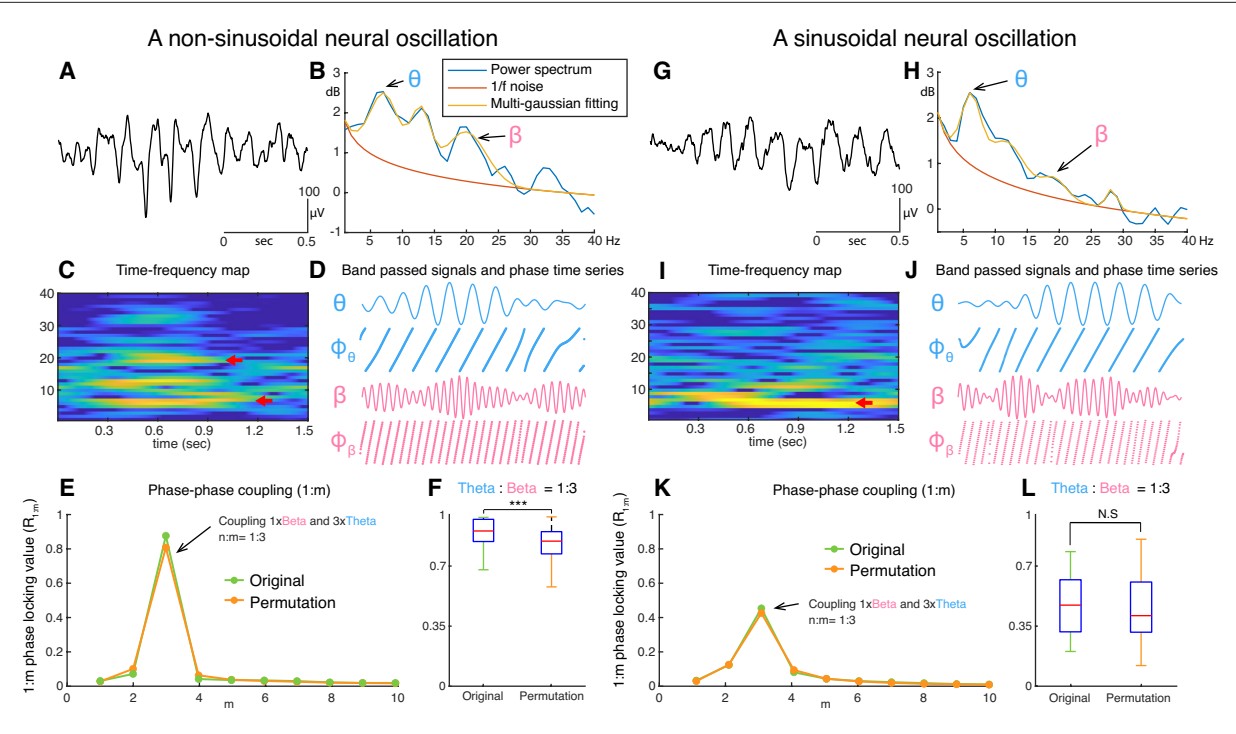

**Figure 1.** Examples of non-sinusoidal and sinusoidal neural oscillations recorded from the human auditory cortex. Detecting the presence, onset/offset, and fundamental frequency of non-sinusoidal oscillations is challenging. This is because the power spectrum of the non-sinusoidal theta-band oscillation (**A**) exhibits multiple harmonic peaks in the alpha and beta bands (**B**). The peaks of these harmonics are also exhibited in the time-frequency domain (**C**). To determine whether these peaks are independent oscillations or harmonics of the fundamental frequency, we tested whether fundamental theta oscillation and potential beta-band harmonic oscillations exhibit a 1:3 phase-locking (**D–F**), i.e., whether the beta-band oscillation is a true third harmonic of the fundamental theta-band oscillation. In our test, we found that the theta-band oscillation was significantly phase-locked to the beta-band oscillation with a 1:3 ratio in their frequencies (**F**, number of permutation = 300, p<0.001). This means that the tested theta- and beta-band oscillations are part of one single non-sinusoidal neural oscillation. We applied the same statistical test to a more sinusoidal neural oscillation (**G**). Since this neural oscillation more closely resembles a sinusoidal shape, it does not exhibit any prominent harmonic peaks in the alpha and beta bands within the power spectrum (**H**) and time-frequency domain (**I**). Consequently, our test found that the phase of the theta-band and beta-band oscillations were not phase-locked (**J–L**). Thus, this statistical test suggests the absence of a harmonic structure.

we introduce the cyclic homogeneous oscillation (CHO) detection method to identify neural oscillations using an autocorrelation analysis to identify whether a neural oscillation is an independent oscillation or a harmonic of another oscillation. Autocorrelation is a statistical measure that assesses the degree of similarity between a time series and a delayed version of itself.

Thus, autocorrelation can explain the periodicity of a signal without assuming that the signal is sinusoidal. Further, the peaks in the output of the autocorrelation function indicate the fundamental frequency of the neural oscillation. As shown in *Figure 2*, irrespective of the shape of neural oscillation (*Figure 2A and C*), the fundamental frequency can be determined from the positive peak-to-peak intervals (see *Figure 2B and D*). Despite autocorrelation being a well-known method to identify the fundamental frequency of a signal, its application to neural oscillations has been impeded by the requirement to accurately determine the onset and offset of the oscillation.

To overcome this limitation, we combine the autocorrelation method with the oscillation event (OEvent) method (*Neymotin et al., 2022*) to determine the onset/offset of oscillations. In this approach, OEvent determines bounding boxes in the time-frequency domain that mark the onset and offset of suspected oscillations. Each bounding box is generated by identifying a period of significantly increased power from averaged power spectrum. To further improve OEvent, we replaced the empirical threshold that identifies bounding boxes in the time-frequency domain with a parametric threshold driven by an estimation of the underlying 1/f noise (*Donoghue et al., 2020*), as shown in *Figure 3A*.

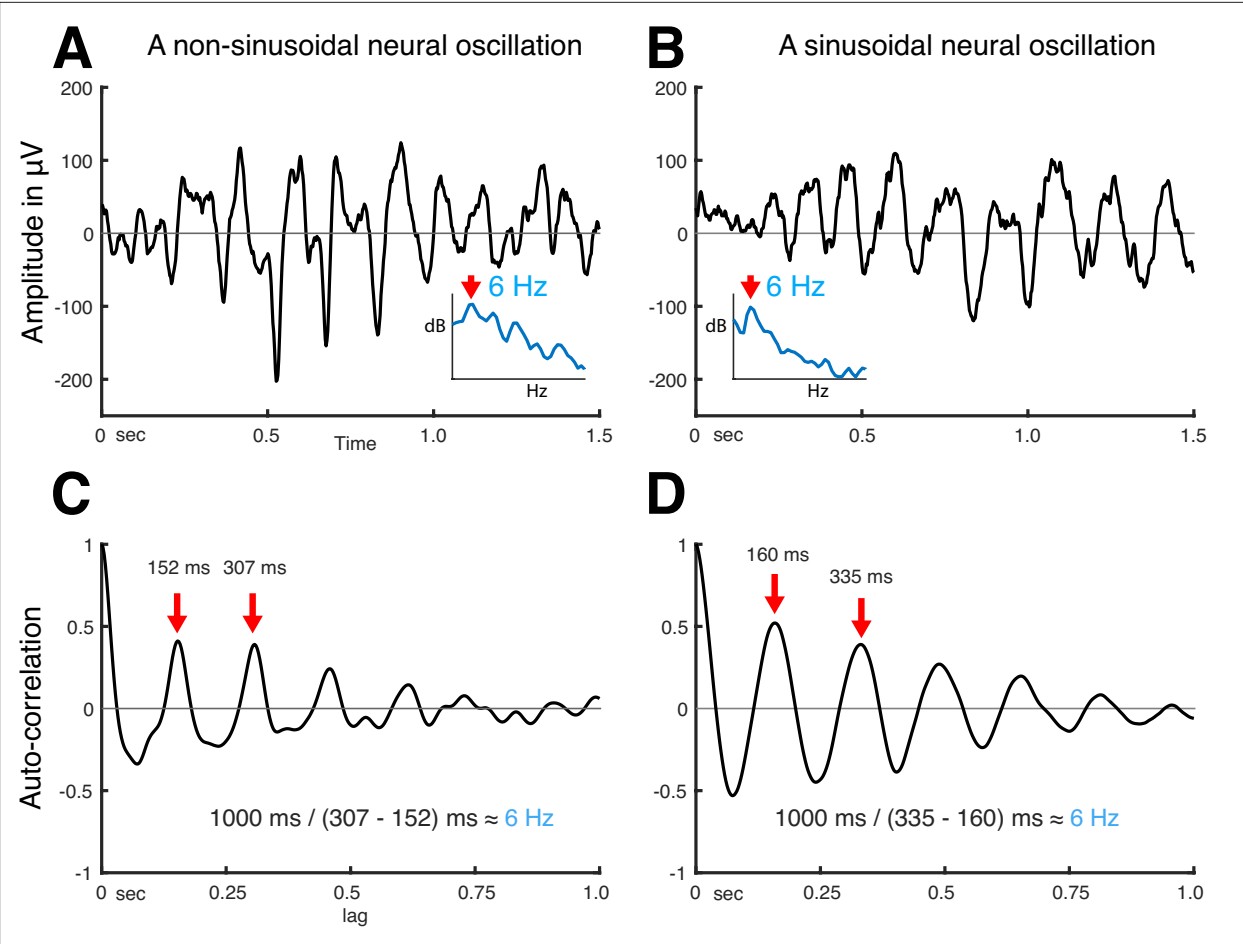

**Figure 2.** Using autocorrelation to determine the fundamental frequency of non-sinusoidal and sinusoidal neural oscillations recorded from the human auditory cortex. (**A**) Temporal dynamics of non-sinusoidal and (**B**) sinusoidal neural oscillation and (**C, D**) their autocorrelation. The periodicity of peaks in the autocorrelation reveals the fundamental frequency of the underlying oscillation. Asymmetry in peaks and troughs of the autocorrelation is indicative of a non-sinusoidal oscillation.

Furthermore, we improved OEvent to reject any short-cycled oscillations that could represent evoked potentials (EPs), event-related potentials (ERPs), or spike activities, as shown in *Figure 3B*. In general, EPs or ERPs in neural signals generate less than 2 cycles of fluctuations. Large-amplitude EPs, ERPs, and spike activities can result in spurious oscillatory power in the frequency domain (*de Cheveigné and Nelken, 2019*; *Donoghue et al., 2020*; *Donoghue et al., 2022*).

In the final step, we determine the oscillation's periodicity and fundamental frequency by identifying positive peaks in the autocorrelation of the signal. As shown for a representative oscillation in *Figure 3C*, the center frequency of the highlighted bounding box is 24 Hz, but the periodicity of the underlying raw signal does not match the calculated fundamental frequency of 7 Hz. Consequently, this bounding box at 24 Hz will be rejected. Finally, we merge those remaining bounding boxes that neighbor each other in the frequency domain and overlap more than 75% (*Neymotin et al., 2022*) in time.

In summary, the presented CHO method identifies neural oscillations that fulfill the following three criteria: (1) oscillations (peaks over $1/f$ noise) must be present in the time and frequency domains; (2) oscillations must exhibit at least 2 full cycles; and (3) oscillations must have autocorrelation. These criteria are supported by studies in the neuroscience literature (*Buzsáki and Draguhn, 2004*; *Niedermeyer and da Silva, 2005*; *Buzsaki, 2006*; *Cohen, 2014*; *de Cheveigné and Nelken, 2019*; *Donoghue et al., 2020*; *Donoghue et al., 2022*). The synthesis of these criteria into the presented method allows us to detect and identify non-sinusoidal oscillations and their fundamental frequency. This is because criteria #1 (i.e. the presence of an oscillation) and #2 (i.e. the length of the oscillation) identify

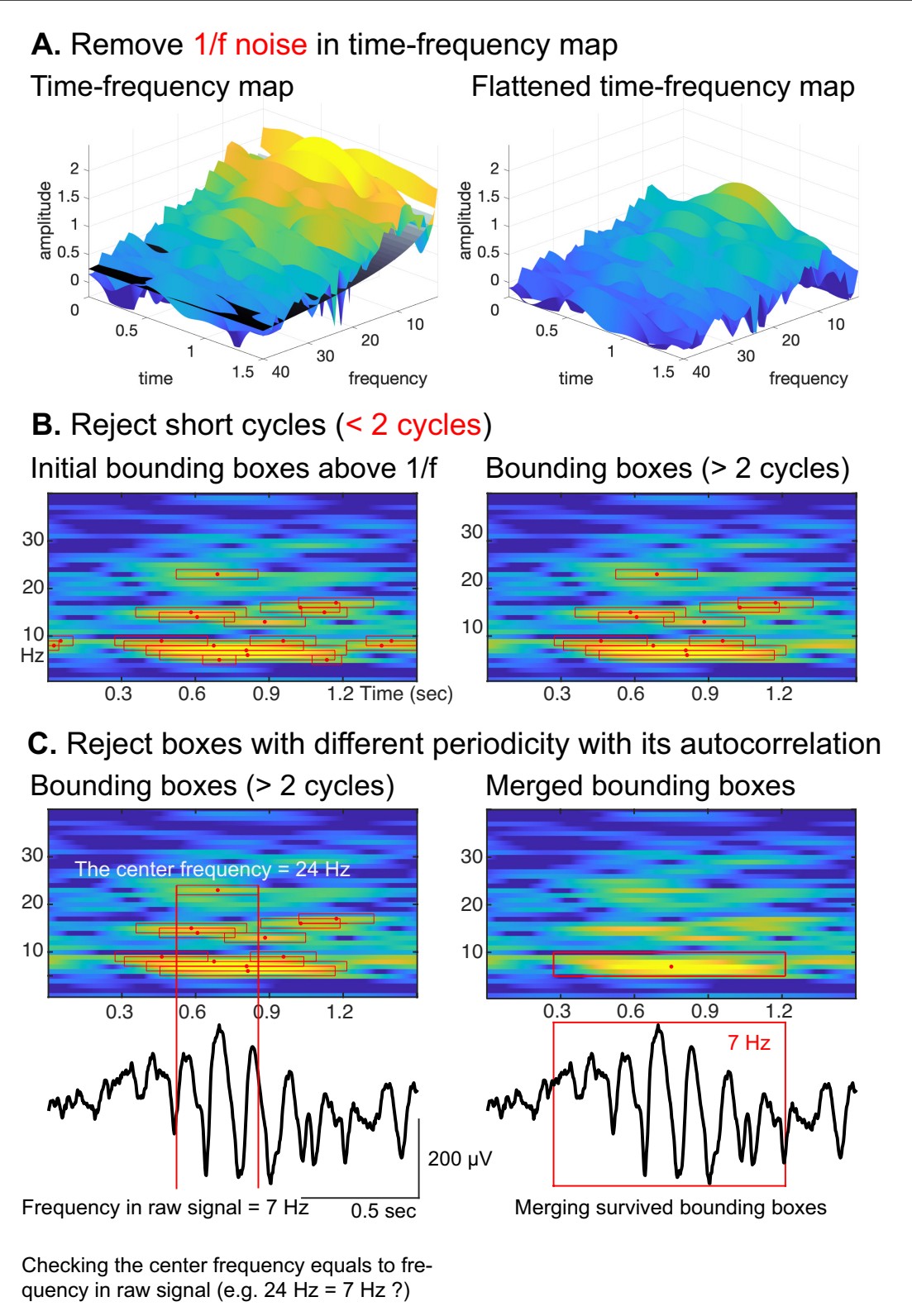

**Figure 3.** Procedural steps of cyclic homogeneous oscillation (CHO). (**A**) First, to identify periodic oscillations, CHO removes the underlying 1/f aperiodic noise in the time-frequency space and generates initial bounding boxes of candidate oscillations. (**B**) In the second step, CHO rejects bounding boxes that exhibit less than 2 oscillatory cycles. (**C**) In the final step, CHO limits the analysis to only those bounding boxes that exhibit the same frequency in the time-frequency map and autocorrelation. Each remaining bounding box is characterized by onset/offset, frequency range, center frequency, and number of cycles.

potential oscillations, which are then tested to be fundamental oscillations using an autocorrelation analysis described in criteria #3 (i.e. the periodicity of an oscillation).

To verify and validate CHO, we applied the above-presented principle criteria on simulated non-sinusoidal signals and human electrophysiological signals, including ECoG signals recorded from the lateral brain surface, electroencephalography (EEG) signals recorded from the scalp, and local field potentials recorded from the hippocampus using stereo EEG (SEEG). We further validated our approach by comparing CHO to other commonly used methods.

To determine the spectral accuracy in detecting the peak frequency of non-sinusoidal oscillations, we compared CHO to established methods, including the fitting of oscillations using 1/*f* (*FOOOF*, also known as *specparam*, ***Donoghue et al., 2020***), the OEvent method (***Neymotin et al., 2022***), and the Spectral Parameterization Resolved in Time (*SPRiNT*, ***Wilson et al., 2022***) methods. Moreover, to determine the spectro-temporal accuracy in detecting both the peak frequency and the onset/offset of non-sinusoidal oscillations, we compared CHO with the *OEvent* method.

The selection of *FOOOF*, *SPRiNT*, and *OEvent* is based on their fundamental approaches. To the best of our knowledge, *FOOOF* is the most representative method for detecting the peak frequency of neural oscillations. *SPRiNT* expands the FOOOF method into the time-frequency domain, and OEvent can determine the onset/offset of the detected oscillations.

## Results

The following sections describe the results of our study: The first section presents simulation results by comparing the accuracy of CHO with that of existing methods in detecting non-sinusoidal oscillations. The second section reports physiological results by comparing the accuracy of CHO with that of established methods in detecting oscillations within in vivo recordings.

### Synthetic results

To determine the specificity and sensitivity of CHO in detecting neural oscillations, we applied CHO to synthetic non-sinusoidal oscillatory bursts (2.5 cycles, 1–3 s long) convolved with 1/*f* noise, also known as pink noise, which has a power spectral density that is inversely proportional to the frequency of the signal. As shown in ***Figure 4***, we generated 5-s-long 1/*f* signals composed of pink noise and added non-sinusoidal oscillations of different lengths (1 cycle, 2.5 cycles, 1 s duration, and 3 s duration). The rightmost panel of ***Figure 4A*** shows two examples of non-sinusoidal oscillations (2.5 cycles and 2 s duration) along with their power spectra. As can be seen in ***Figure 4A***, longer non-sinusoidal oscillations exhibit stronger harmonic peaks.

Our results in ***Figure 4B–D*** demonstrate that CHO outperforms conventional techniques in specificity and accuracy for detecting the peak frequency of non-sinusoidal oscillations. High specificity depends on high true-negative and low false-positive rates. For conventional methods, we expected harmonic oscillations to increase the false-positive rate and one-cycled oscillations to decrease the true-negative rate. As expected, conventional methods detected harmonic and one-cycled oscillations as true oscillations. For example, the average specificity of SPRiNT was below 0.3, which was significantly lower than the robust specificity of CHO across the entire range of signal-to-noise ratio (SNR).

We also observed that CHO requires a higher SNR to detect the presence of oscillations. Sensitivity depends on the true-positive and the false-negative rates. We found existing methods to be overly sensitive in detecting the presence of oscillations. At the same time, this severely limits their specificity and, thus, their ability to accurately detect the presence and frequency of an oscillation. Based on our physiological datasets, we found the average SNR of oscillations in EEG and ECoG to be –7 and –6 dB, respectively (***Figure 4—figure supplement 1***). When tested at these physiologically motivated SNR levels, we found that the sensitivity of CHO is comparable to that of SPRiNT. Overall, when considering the accuracy combined with specificity and sensitivity, CHO outperformed all other methods in detecting the peak frequency of non-sinusoidal oscillations at the physiologically motivated SNR levels.

In addition to determining the accuracy in detecting the presence of oscillations and determining their peak frequency, we also determined the accuracy of all methods in detecting the onset and offset of oscillations. This comparison is limited to OEvent because FOOOF and SPRiNT methods

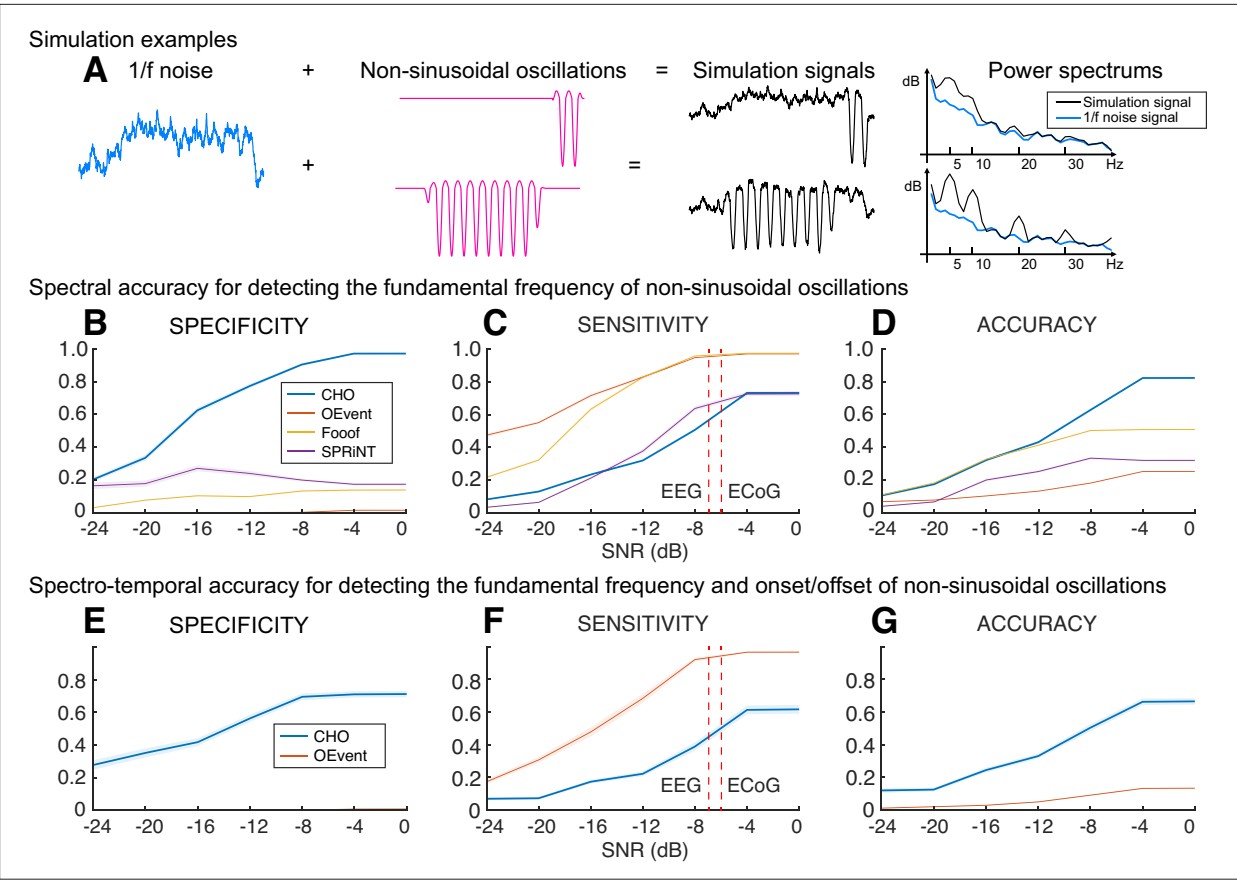

**Figure 4.** Performance of cyclic homogeneous oscillation (CHO) in detecting synthetic non-sinusoidal oscillations. (**A**) We evaluated CHO by verifying its specificity, sensitivity, and accuracy in detecting the fundamental frequency of non-sinusoidal oscillatory bursts (2.5 cycles, 1–3 s long) convolved with 1/*f* noise. (**B–D**) CHO outperformed existing methods in detecting the fundamental frequency of non-sinusoidal oscillation (FOOOF: fitting of oscillations using 1/*f* ***Donoghue et al., 2020***, oscillation event (OEvent) ***Neymotin et al., 2022***: Oscillation event detection method, and SPRiNT ***Wilson et al., 2022***: Spectral Parameterization Resolved in Time) in specificity and accuracy, but not in sensitivity. CHO exhibited fewer false-positive and more true-negative detections than existing methods. (**C**) However, at signal-to-noise ratio (SNR) levels of alpha oscillations found in electroencephalographic (EEG) and electrocorticographic (ECoG) recordings (i.e. –7 and –6 dB, respectively), the sensitivity of CHO in detecting the peak frequency of non-sinusoidal oscillation is comparable to that of SPRiNT. (**D**) This means that the overall accuracy of CHO was higher than that of existing methods. (**E–G**) CHO outperformed existing methods in detecting the fundamental frequency and onset/offset of non-sinusoidal oscillation. (**F**) Similar to the results shown in (**C**) CHO can effectively detect the fundamental frequency and onset/offset for more than half of all oscillations at SNR levels of alpha oscillations found in EEG and ECoG recordings.

The online version of this article includes the following figure supplement(s) for figure 4:

**Figure supplement 1.** Signal-to-noise ratio (SNR) histograms of (**A**) electroencephalography (EEG) and (**B**) electrocorticography (ECoG).

**Figure supplement 2.** Synthetic sinusoidal oscillations.

cannot determine the onset and offset of short oscillations. In this analysis, CHO outperformed the OEvent method in specificity but not sensitivity, as shown in ***Figure 4E–G***. Specifically, we found performance trends similar to those in our previous simulation result (***Figure 4B–D***). Thus, CHO outperforms conventional techniques in specificity for detecting both the peak frequency and onset/offset of oscillations.

## Empirical results

We further assessed CHO by testing it on electrophysiological signals recorded from human subjects. Specifically, we evaluated CHO on ECoG (x1–x8, 8 subjects) and EEG (y1–y7, 7 subjects) signals recorded during the pre-stimulus period of an auditory reaction-time task. Furthermore, we also evaluated CHO on signals recorded during resting state from cortical areas and hippocampus using ECoG (ze1–ze8, 6 subjects) and SEEG (zs1–zs6, 6 subjects).

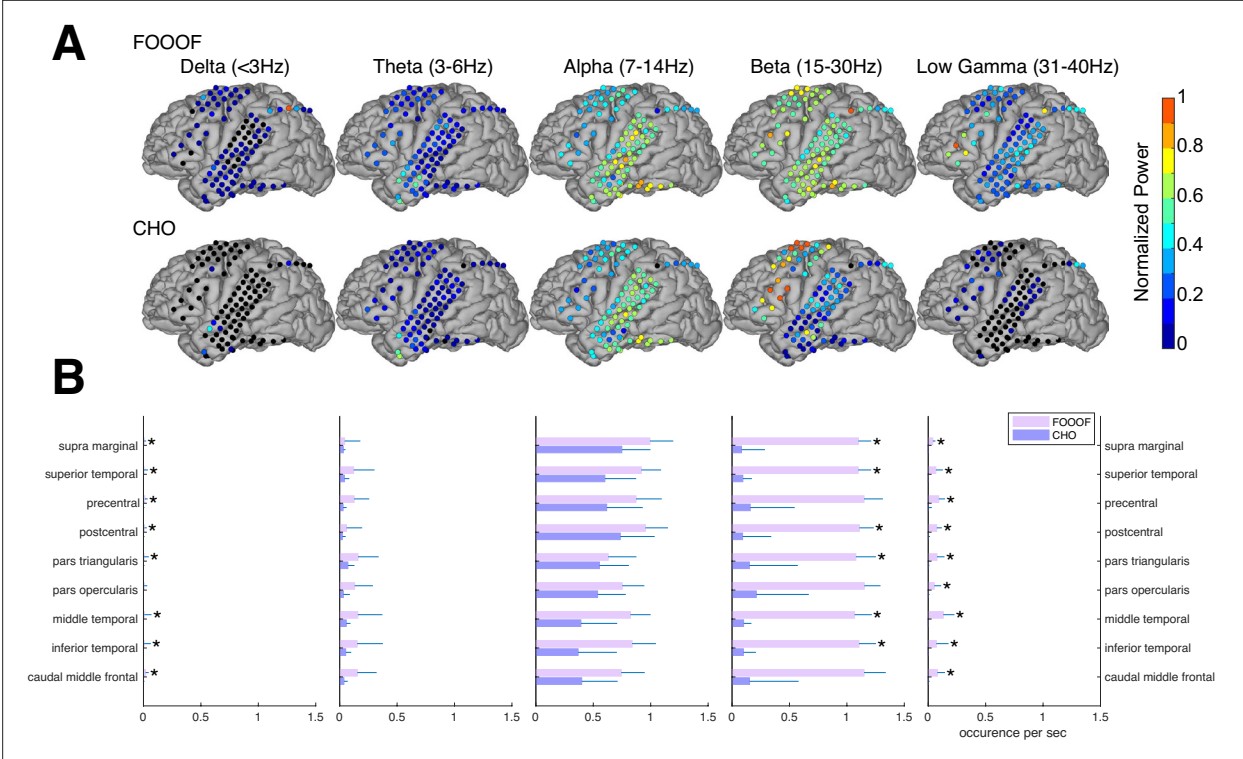

**Figure 5.** Validation of cyclic homogeneous oscillation (CHO) in detecting oscillations in electrocorticographic (ECoG) signals. (**A**) We applied CHO and fitting of oscillations using 1/f (FOOOF) to determine the fundamental frequency of oscillations from ECoG signals recorded during the pre-stimulus period of an auditory reaction-time task. FOOOF detected oscillations primarily in the alpha- and beta-band over STG and pre-motor area. In contrast, CHO also detected alpha-band oscillations primarily within STG, and more focal beta-band oscillations over the pre-motor area, but not STG. (**B**) We investigated the occurrence of each oscillation within defined cerebral regions across eight ECoG subjects. The horizontal bars and horizontal lines represent the median and median absolute deviation (MAD) of oscillations occurring across the eight subjects. An asterisk (*) indicates statistically significant differences in oscillation detection between CHO and FOOOF (Wilcoxon rank-sum test, n=8, p<0.05 after Bonferroni correction).

The online version of this article includes the following figure supplement(s) for figure 5:

**Figure supplement 1.** Electrocorticographic (ECoG) results using fitting of oscillations using 1/f (FOOOF) and cyclic homogeneous oscillation (CHO) for all subjects.

## ECoG results

In the auditory reaction-time task, we expected to observe neural low-frequency oscillations during the pre-stimulus period within task-relevant areas, such as the auditory and motor cortex. As we expected, we found alpha and beta oscillations within these cortical areas. We compared the topographic distribution of the oscillations detected by FOOOF with those detected by CHO. As shown in *Figure 5A* for one representative subject, FOOOF detected the presence of alpha and beta oscillations within temporal and motor cortex. In contrast, while CHO also detected alpha oscillations in temporal and motor cortex, it only detected beta oscillations in motor cortex. We found this pattern to be consistent across subjects, as shown in *Figure 5B* and *Figure 5—figure supplement 1*.

We compared neural oscillation detection rates between CHO and FOOOF across eight ECoG subjects. We used FreeSurfer (*Fischl, 2012*) to determine the associated cerebral region for each ECoG location. Each subject performed approximately 400 trials of a simple auditory reaction-time task. We analyzed the neural oscillations during the 1.5-s-long pre-stimulus period within each trial. CHO and FOOOF demonstrated statistically comparable results in the theta and alpha bands despite CHO exhibiting smaller median occurrence rates than FOOOF across eight subjects. Notably, within the beta band, excluding specific regions such as precentral, pars opercularis, and caudal middle frontal areas, CHO's beta oscillation detection rate was significantly lower than that of FOOOF (Wilcoxon rank-sum test, p<0.05 after Bonferroni correction). This suggests comparable detection rates between CHO and FOOOF in pre-motor and Broca's areas, while the detection of beta oscillations by FOOOF

in other regions, such as the temporal area, may represent harmonics of theta or alpha, as illustrated in *Figure 5A and B*. Furthermore, FOOOF exhibited a higher sensitivity in detecting delta, theta, and low gamma oscillations overall, although both CHO and FOOOF detected only a limited number of oscillations in these frequency bands.

## EEG results

We expected that the EEG would exhibit similar results as seen in the ECoG results. Indeed, the EEG results mainly exhibit alpha and beta oscillations during the pre-stimulus periods of the auditory reaction-time task, as shown in *Figure 6*. Specifically, FOOOF found alpha oscillations in mid-frontal and visual areas and beta oscillations throughout all areas of the scalp. In contrast, CHO found more focal visual alpha and pre-motor beta. Furthermore, the low gamma oscillations detected by CHO were also more focal than those detected by FOOOF. We found these results to be consistent across subjects (see *Figure 6B and C* and *Figure 6—figure supplement 1*).

We assessed the difference in neural oscillation detection performance between CHO and FOOOF across seven EEG subjects. We used EEG electrode locations according to the 10–10 electrode system (*Nuwer, 2018*) and assigned each electrode to the appropriate underlying cortex (e.g. O1 and O2 for the visual cortex). Each subject performed 200 trials of a simple auditory reaction-time task. We analyzed the neural oscillations during the 1.5-s-long pre-stimulus period. In the alpha-band, CHO and FOOOF presented statistically comparable outcomes. However, CHO exhibited a greater alpha detection rate for the visual cortex than for the pre-motor cortex, as shown in *Figure 6B and C*. The entropy of CHO's alpha oscillation occurrences (3.82) was lower than that of FOOOF (4.15), with a maximal entropy across 64 electrodes of 4.16. Furthermore, in the beta band, CHO's entropy (4.05) was smaller than that of FOOOF (4.15). These findings suggest that CHO may offer a more region-specific oscillation detection than FOOOF. As illustrated in *Figure 6C*, CHO found fewer alpha oscillations in pre-motor cortex (FC2 and FC4) than in occipital cortex (O1 and O2), while FOOOF found more beta oscillation occurrences in pre-motor cortex (FC2 and FC4) than in occipital cortex. However, FOOOF found more alpha and beta oscillations in visual cortex than in pre-motor cortex. Consistent with ECoG results, FOOOF demonstrated heightened sensitivity in detecting delta, theta, and low gamma oscillations. Nonetheless, both CHO and FOOOF identified only a limited number of oscillations in delta and theta frequency bands. Contrary to the ECoG results, FOOOF found more low gamma oscillations in EEG subjects than in ECoG subjects.

## Onset and offset of neural oscillations

So far, we have established that CHO can localize beta rhythms within pre-motor cortex in EEG and ECoG. Here, we are interested in determining the accuracy of the onset/offset detection of neural oscillations. For this purpose, we tested whether CHO, applied to signals recorded from auditory cortex during an auditory reaction-time task, can accurately detect the transition between resting and task periods. Specifically, we expected CHO to detect the offset times of neural oscillations after the stimulus onset (i.e. a beep tone that remained until a button was pressed). Based on the principle of event-related de-/synchronization (*Pfurtscheller and Lopes da Silva, 1999*), cortical neurons may be de-synchronized to process an auditory stimulus. As shown in *Figure 7*, CHO successfully detected offset times of 7 Hz neural oscillations. During the pre-stimulus period, the distribution of the onset time remains uniform, reflecting the subject waiting for the stimulus. In contrast, after the stimulus onset, the distribution of onset times becomes Gaussian, reflecting the variable reaction time to the auditory stimulus. Of note, the detection of onset times peaks 950 ms post-stimulus, which occurs significantly later than the button press that happens 200 ms post-stimulus (*Figure 7B*).

Similar to the distribution of onset times, the distribution of offset times remained uniform throughout the pre-stimulus period. After stimulus onset, the distribution becomes Gaussian, with a peak of offset detections at 300 ms post-stimulus, or 200 ms post-response (i.e. the button press) (*Figure 7C*).

In summary, this means that, on average, the detected 7 Hz oscillations de-synchronized 250 ms and synchronized 900 ms, post-stimulus, respectively.

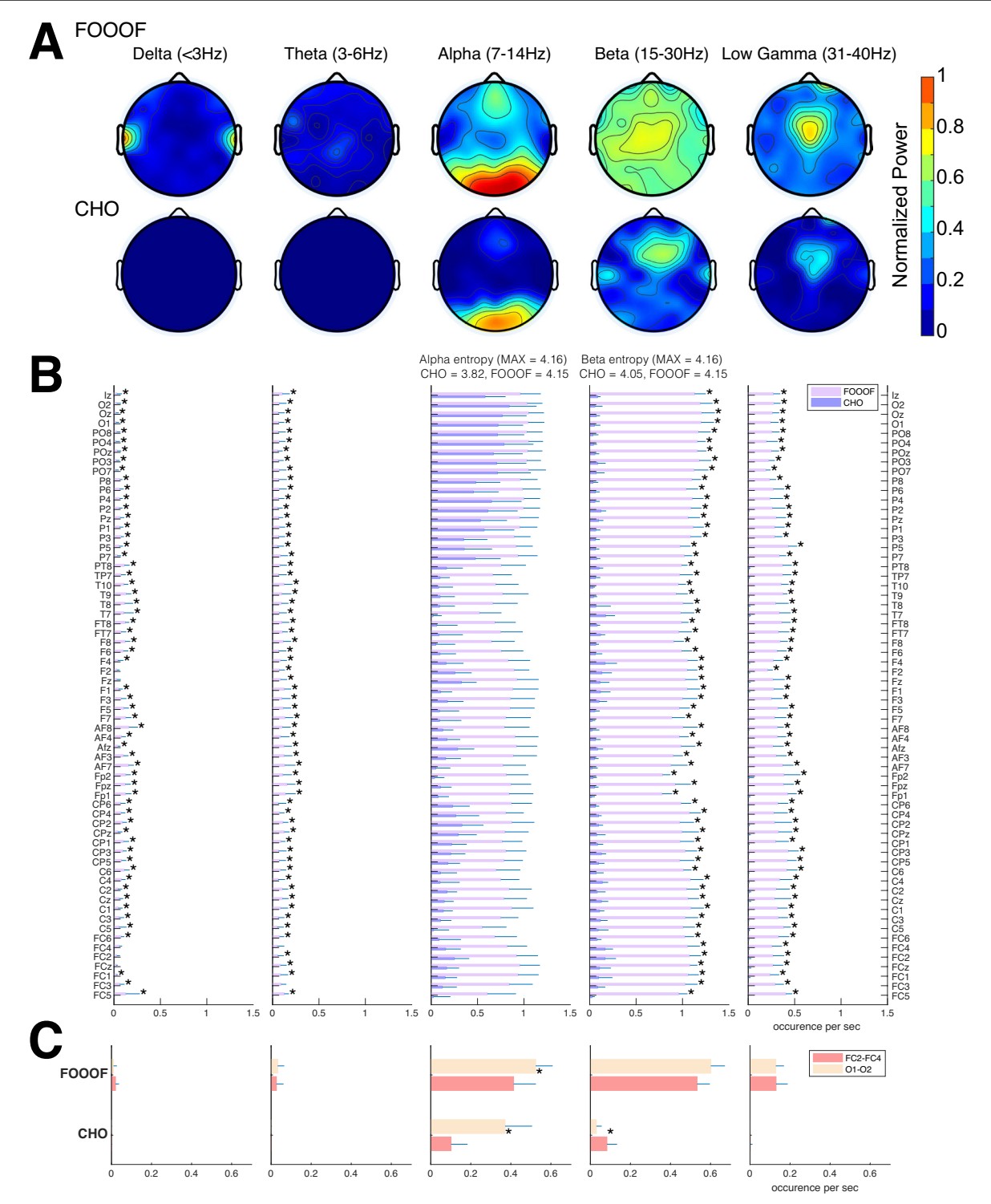

**Figure 6.** Validation of cyclic homogeneous oscillation (CHO) in detecting oscillations in electroencephalographic (EEG) signals. (**A**) We applied CHO and fitting of oscillations using 1/$f$ (FOOOF) to determine the fundamental frequency of oscillations from EEG signals recorded during the pre-stimulus period of an auditory reaction-time task. FOOOF primarily detected alpha-band oscillations over frontal/visual areas and beta-band oscillations across all areas (with a focus on central areas). In contrast, CHO detected alpha-band oscillations primarily within visual areas and detected more focal beta-band oscillations over the pre-motor area, similar to the electrocorticographic (ECoG) results shown in *Figure 5*. (**B**) We investigated the occurrence of each oscillation within the EEG signals across seven subjects. An asterisk (*) indicates statistically significant differences in oscillation detection between CHO and FOOOF (Wilcoxon rank-sum test, n=7, p<0.05 after Bonferroni correction). CHO exhibited lower entropy values of alpha and beta occurrence

*Figure 6 continued on next page*

*Figure 6 continued*

than FOOOF across 64 channels. (**C**) We compared the performance of FOOOF and CHO in detecting oscillation across visual and pre-motor-related EEG channels. CHO detected more alpha and beta oscillations in visual cortex than in pre-motor cortex. FOOOF detected alpha and beta oscillations in visual cortex than in pre-motor cortex.

The online version of this article includes the following figure supplement(s) for figure 6:

**Figure supplement 1.** All electroencephalographic (EEG) results using fitting of oscillations using 1/*f* (FOOOF) and cyclic homogeneous oscillation (CHO).

## SEEG results

We also investigated neural oscillations within the hippocampus. Specifically, we were interested in the frequency and duration of hippocampal oscillations, which are known to be non-sinusoidal and a hallmark of memory processing (*Buzsaki, 2006*; *Lundqvist et al., 2016*). Using the CHO method, we plotted a representative example of detected hippocampal fast theta bursts (*Lega et al., 2012*; *Goyal et al., 2020*), as shown in *Figure 8*. As expected, the non-sinusoidal alpha-band oscillations also resulted in harmonic oscillations in the beta band, which, while not clearly visible in the power spectrum (*Figure 8B*), can be clearly seen in the time-frequency analysis (*Figure 8D* and *Figure 8E*). In contrast to the ECoG and EEG results, the frequency of beta-band oscillations in the hippocampus exhibited a frequency close to the alpha-band (7–14 Hz). CHO found primarily alpha-band oscillations in the hippocampus (see *Figure 8—figure supplement 1*). When comparing the consistency between CHO and FOOOF across hippocampal locations, CHO exhibits more specific results with less overlap between alpha and beta locations and almost no detection in the low gamma band (30–40 Hz). For example, subject zs4 in *Figure 8—figure supplement 1* shows alpha and beta locations mutually supplement each other when using CHO but not when using the FOOOF method. However, we did not find a statistically significant difference between CHO and FOOOF due to the small number of subjects and variable electrode locations within hippocampus across the six SEEG subjects.

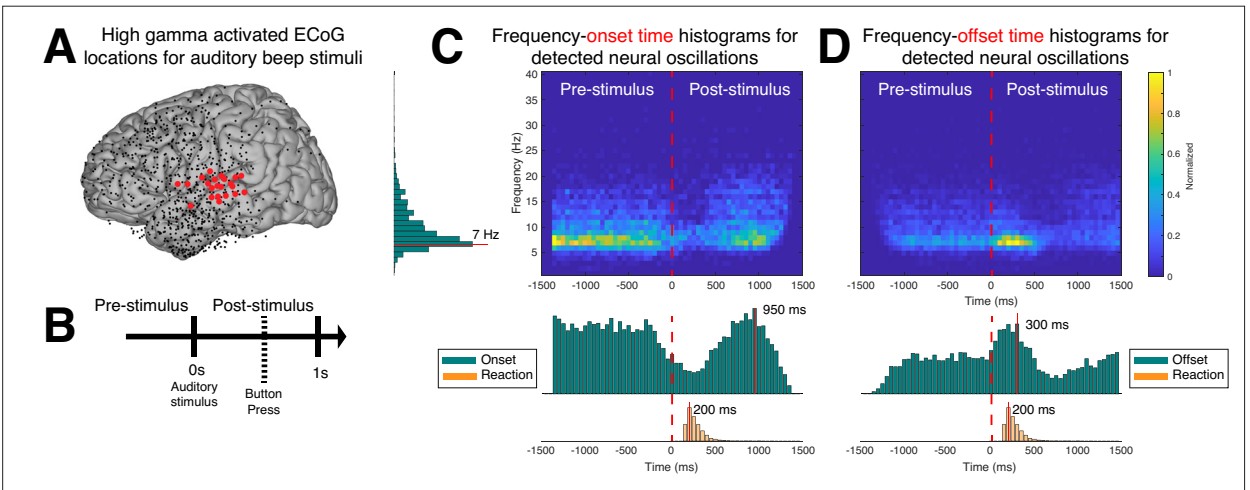

**Figure 7.** Application of cyclic homogeneous oscillation (CHO) in determining the spatiotemporal characteristics of neural oscillations in electrocorticographic (ECoG) signals during a reaction-time task. (**A**) We selected those cortical locations (red) from all locations (black) that exhibited a significant broadband gamma response to an auditory stimulus in a reaction-time task. (**B**) In this task, the subjects were asked to react as fast as possible with a button press to a salient auditory stimulus. (**C–D**) Onset and offset times of detected neural oscillations. Fundamental oscillations were centered around 7 Hz (left histogram). Onset and offset times during pre-stimulus period exhibited a uniform distribution, indicating that 7 Hz oscillations randomly started and stopped during this period. A trough in the onset and a peak in the offset of 7 Hz oscillations is visible from the histograms, indicating a general decrease of the presence of neural oscillations immediately following the auditory stimulus. The subjects responded with a button press within 200 ms of the auditory stimulus, on average. The prominent peak in the offset and onset of oscillations at 300 and 950 ms post-stimulus, respectively, indicates a suspension of oscillations in response to the auditory stimulus, and their reemergence after the execution of the button press behavior.

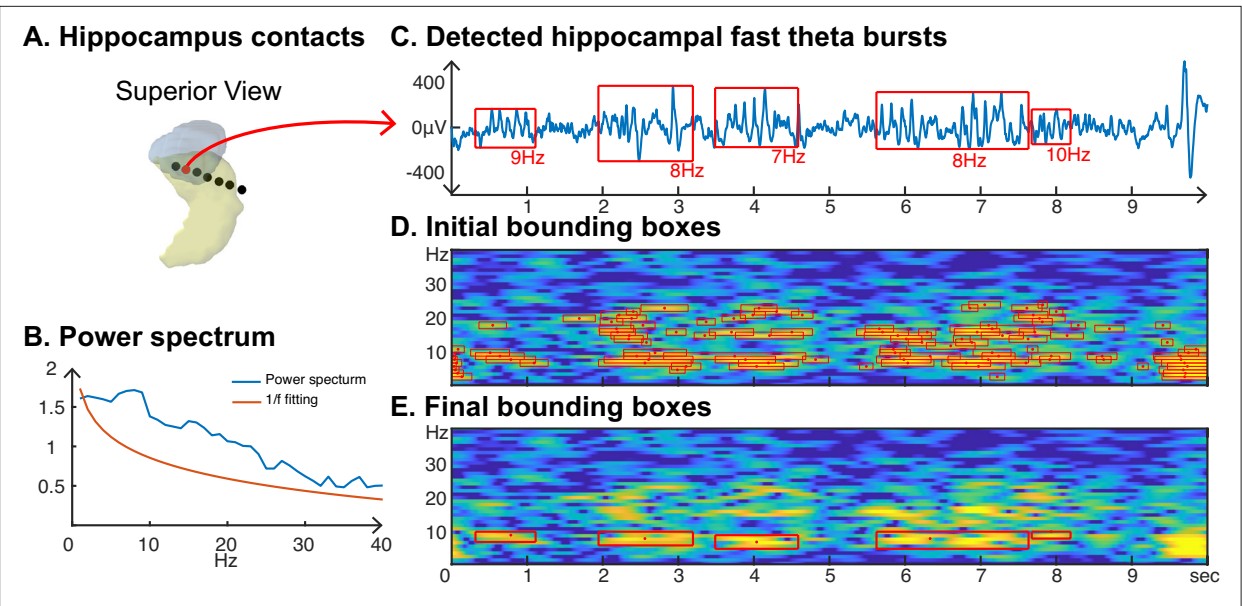

**Figure 8.** Application of cyclic homogeneous oscillation (CHO) in determining the fundamental frequency and duration of hippocampal oscillations in stereoelectroencephalographic (SEEG) signals during resting state. (**A**) We recorded hippocampal oscillations from one representative human subject implanted with SEEG electrodes within the left anterior hippocampus. (**B**) Power spectrum (blue) and 1/f trend (red) for one electrode within the anterior-medial left hippocampus (red dot in A). The power spectrum of a 10-s-long hippocampal signal indicates the presence of neural activity over a 1/f trend across a wide frequency band up to 30 Hz. (**C**) In marked contrast to the relatively unspecific results indicated by the power spectrum, CHO detected several distinct hippocampal fast theta bursts. (**D**) This detection is based on first denoising the power spectrum using 1/f fitting (principle criterion #1 of CHO), which yields initial bounding boxes that include short-cycled oscillations and harmonics. (**E**) The autocorrelation step then successfully removes all short-cycled oscillations and harmonics, with only those bounding boxes remaining that exhibit a fundamental frequency.

The online version of this article includes the following figure supplement(s) for figure 8:

**Figure supplement 1.** All results from six stereoelectroencephalographic (SEEG) subjects using the fitting of oscillations using 1/f (FOOOF) and cyclic homogeneous oscillation (CHO) methods.

## Frequency and duration of neural oscillations

Here, we are interested in identifying the predominant frequency and duration of neural oscillations for specific brain areas during the resting state. For this purpose, we first determined the specific Brodmann area of each recording electrode using an intracranial electrode localization tool, Versatile Electrode Localization Framework (**Adamek, 2022**). Next, we investigated electrodes belonging to the primary auditory cortex (i.e. Brodmann areas 41 and 42), as shown in **Figure 9A**. We found that 7 and 11 Hz oscillations were the predominant neural oscillations for electrodes near the primary auditory cortex. The average duration of an 11 Hz oscillation was 450 ms. Next, our results for primary motor cortex (i.e. Brodmann area 4) showed that 7 Hz was the predominant oscillation frequency in the motor cortex with 450 ms duration on average, as shown in **Figure 9B**. We found that motor cortex exhibits more beta-band oscillations (around 500 ms duration) than the auditory cortex. Next, Broca's area exhibited characteristics similar to those of the motor cortex, however, with a predominant beta-band frequency of 17 Hz, which is lower than the 22 or 24 Hz oscillations found in the motor cortex (**Figure 9C**). Lastly, using SEEG electrodes, we investigated neural oscillations within the human hippocampus (**Figure 9D**). This analysis showed that 8 Hz was the predominant oscillatory frequency in the hippocampus with a 450 ms duration on average. During the resting state, neural alpha- and beta-band oscillations within the hippocampus were shorter than in the motor cortex (p<0.05, Wilcoxon rank-sum test, N=6).

## Discussion

Our novel CHO method demonstrates high precision and specificity in detecting neural oscillations in time and frequency domains. The method's specificity enables the detailed study of spatiotemporal

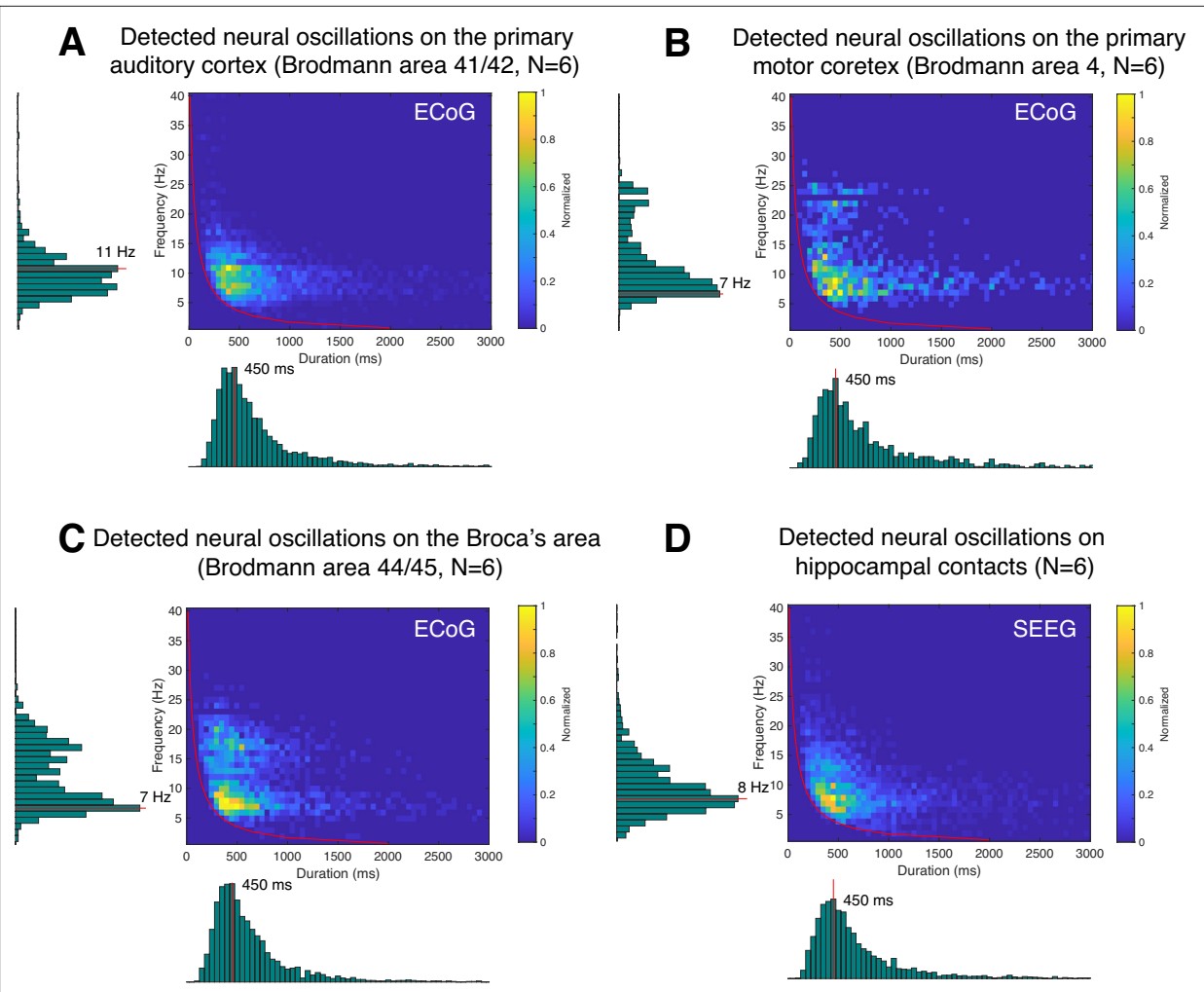

**Figure 9.** Application of cyclic homogeneous oscillation (CHO) in determining the fundamental frequency and duration of neural oscillations in auditory cortex, motor cortex, Broca's area, and hippocampus during resting state. This figure presents the distribution of detected oscillations in a two-dimensional frequency/duration histogram and projected onto frequency and duration axes. The red line indicates the rejection line (less than 2 cycles). (**A**) In primary auditory cortex (Brodmann area 41/42), the most dominant frequency and duration in the auditory cortex was 11 Hz with 450 ms duration. (**B**) The primary motor cortex's most dominant frequency was 7 Hz with 450 ms duration, but more beta rhythms were detected with >500 ms duration than in auditory cortex. (**C**) Broca's area exhibits similar characteristics to that of motor cortex, but dominant beta-band oscillations were found to be less present than in motor cortex. (**D**) Hippocampus primarily exhibits 8 Hz oscillations with 450 ms duration.

dynamics of oscillations throughout the brain and the investigation of oscillatory biomarkers that index functional brain areas.

## High specificity for detecting neural oscillations

In our simulation study, CHO demonstrated high specificity in detecting both the peak and onset/offset of neural oscillations in time and frequency domains. This high specificity directly results from the three criteria we established in this study. The first criterion was that neural oscillations (peaks over 1/*f* noise) must be present in the time and frequency domain. The 1/*f* trend estimation served as a threshold to reject aperiodic oscillatory power in the neural signals (***Donoghue et al., 2020***).

Next, the second condition was that oscillations must exhibit at least 2 complete cycles. This condition distinguishes periodic oscillations from EPs/ERPs and spike artifacts. EPs/ERPs have spectral characteristics that are similar to those of theta or alpha frequency oscillations. To discriminate EP/ERPs from genuine oscillations, we reject them if they don't exhibit peaks over 1/*f* or if they have fewer than two cycles.

The third and final condition is that oscillations should share the same periodicity as their autocorrelation. This is because positive peaks in the autocorrelation can identify the oscillation's fundamental frequency even if it is non-sinusoidal. The bounding boxes help us to identify possible onsets/offsets of neural oscillations. Moreover, calculating the autocorrelation of the raw signals within a bounding box provides the true periodic frequency of the raw signal. We then reject any bounding boxes for which the periodicity of the raw signal is not in alignment with the true periodic frequency revealed by the autocorrelation. This third condition is important in rejecting harmonic peaks over $1/f$ noise in the frequency domain. Furthermore, it is also effective in rejecting spurious oscillations, which are broadly generated by spike activities in the frequency domain (*de Cheveigné and Nelken, 2019*).

To calculate the autocorrelation, we first needed to determine the onset/offset of the potential oscillations. The first and second criteria serve as a triage in finding the onset/offset of genuine oscillations. Thus, these three principle criteria were essential to reject aperiodic harmonic oscillations and increase CHO's specificity in detecting both the peak frequency and the onset/offset of non-sinusoidal oscillations. We also evaluated CHO on purely sinusoidal oscillations (see *Figure 4—figure supplement 2*). The results of this analysis show that even in the absence of any asymmetry in the oscillations, CHO still outperforms existing methods in specificity. It further shows that the sensitivity increases with increasing SNR. Even though this analysis is based on synthetic sinusoidal oscillations, our results demonstrated that existing methods are susceptible to noise which results in the detection of spurious oscillations. However, as expected, both FOOOF and SPRiNT methods exhibited reasonable specificity when applied to sinusoidal signals.

## Focal localization of beta oscillations

Beta oscillations occur within the 13–30 Hz band throughout various brain regions, including the motor cortex. In the motor cortex, beta oscillations are thought to be involved in motor planning and execution. Studies have shown that beta oscillations increase and decrease in power during movement preparation and movement execution, respectively (*Pfurtscheller and Lopes da Silva, 1999*; *Jenkinson and Brown, 2011*; *Doyle et al., 2005*; *Senkowski et al., 2006*). In our empirical results based on the presented ECoG dataset, CHO found focal beta oscillations to occur within pre-motor and frontal cortex prior to the button response, as shown in *Figure 5*. These findings were consistent across subjects. Conventional methods found alpha and beta oscillations in the auditory cortex, while CHO found only select beta oscillations. This suggests that most of the beta oscillations detected by conventional methods within auditory cortex may be simply harmonics of the predominant asymmetric alpha oscillation. Along the same line, conventional methods found beta and low gamma oscillations in pre-motor and frontal areas, while CHO found predominantly beta oscillations. This suggests that low gamma oscillations detected by conventional methods are harmonics of beta oscillations.

In the EEG results, CHO found focal visual alpha and motor beta oscillations, while the FOOOF found frontal and visual alpha and beta oscillations across broad scalp areas, as shown in *Figure 6*. In contrast to the ECoG results, neither CHO nor FOOOF found auditory alpha oscillations within the temporal areas. This is interesting as FOOOF exhibits a better sensitivity than CHO and suggests that auditory alpha rhythms may be difficult to observe in EEG. Similar to the ECoG results, our analysis confirmed that non-sinusoidal alpha and beta oscillations generate harmonic oscillations in both beta and low gamma in EEG. This shows that our CHO method, which has a high specificity, can detect focal motor beta oscillations.

## Harmonic oscillations in human hippocampus

Recent studies suggest that the frequency range of hippocampal oscillations is wider than previously assumed (<40 Hz in *Cole and Voytek, 2019* or 3–12 Hz in *Li et al., 2022*) and that it does not match the conventional frequency range of theta/alpha rhythms (*Buzsaki, 2006*). This realization stems from the recognition that neural oscillations are non-sinusoidal, and thus require a wide frequency band to be fully captured (*Cole and Voytek, 2019*; *Donoghue et al., 2022*). Adopting a wider frequency band provides more frequency options in fitting the non-sinusoidal shape of brain waves. The recognition of the need to expand the frequency band within oscillation analysis is not limited to the hippocampus. Our ECoG and EEG results show that harmonics can occur in any brain area and frequency band because neural oscillations are inherently non-sinusoidal. A recent study showed that the phase of wideband oscillations could better predict neural firing (*Davis et al., 2020*).

CHO can determine the fundamental frequency of non-sinusoidal oscillations when applied within a wideband analysis, as shown in *Figure 8E*. Moreover, CHO provides onset/offset and the frequency range of an oscillation, allowing us to investigate non-sinusoidal features, such as the degree of asymmetry and amplitudes of troughs/peaks (*Cole and Voytek, 2019*).

## Identifying onset/offset of neural oscillations and its application

Although the frequency of neural oscillations have been extensively investigated, the onset/offset and duration of neural oscillations have remained elusive. Using CHO, the onset/offset and duration of neural oscillations can be revealed, as shown in *Figure 7* and *Figure 9*. Knowing the onset/offset and duration of a neural oscillation is essential for realizing closed-loop neuromodulation. This is because neuromodulation may be most efficient when electrical stimulation is delivered phase-locked to the underlying ongoing oscillation (*Chen et al., 2013*; *Cagnan et al., 2017*; *Cagnan et al., 2019*; *Zanos et al., 2018*; *Shirinpour et al., 2020*). For example, deep-brain stimulation in phase with ongoing oscillation can reduce the stimulation necessary to achieve the desired therapeutic effect (*Cagnan et al., 2017*; *Cagnan et al., 2019*). This improved efficiency in delivering the stimulation therapy reduces power consumption and thus enhances the battery life of the implanted system (*Chen et al., 2013*). Longer battery life means fewer battery changes (which require surgical procedures), or for rechargeable systems, fewer recharging sessions (which require the user's attention). Realizing phase-locked neuromodulation requires detecting the duration of an ongoing oscillation with high specificity and delivering the electrical stimulation at a predicted oscillation phase. The detection and identification with high specificity thus enable neuromodulation applications that depend on phase-locked electrical stimulation.

Moreover, the temporal precision of CHO in detecting neural oscillations can improve the effectiveness of neurofeedback-based systems. For example, a neurofeedback system may provide targeted feedback on the magnitude of the user's alpha oscillation to improve attention and in turn improve task performance. For this purpose, the system must detect the frequency, onset/offset, and duration of the user's alpha oscillation with high specificity. High specificity requires distinguishing other oscillations and artifacts from true physiological alpha-band oscillations. The identification of true neural oscillations with the high specificity of CHO thus enables targeted neurofeedback applications to enhance or restore task performance.

## Illuminating the when, where, what, why, how, and whom of neural oscillations

In our study, we focused on the temporal dynamics ('when'), spatial distribution ('where'), and fundamental frequency ('what') of neural oscillations. However, fully understanding the role of neural oscillations in cognition and behavior also requires investigating their underlying mechanisms ('how'), functional purpose ('why'), and pathologies ('whom').

### Temporal dynamics – the 'when'

CHO demonstrated high specificity in detecting the onset and offset of fundamental non-sinusoidal oscillations (see *Figure 4E*). Using CHO, our study revealed the temporal dynamics of oscillations within the temporal lobe in an auditory reaction-time task. We identified the onsets and offsets of 7 Hz oscillations and, thus, the boundaries in oscillatory activity between resting and task engagement. Our results show a rapid decrease in oscillatory activity for the duration of the auditory stimulus, followed by a rapid reemergence of the oscillatory activity following the cessation of the auditory stimulus (see *Figure 7C and D*). These results shed light on the temporal dynamics of neural oscillatory activity in cognitive processes and how the brain adapts to environmental stimuli.

### Spatial distribution – the 'where'

CHO revealed the spatial distribution of neural oscillations in EEG, SEEG, and ECoG recordings. The spatial distribution of fundamental neural oscillations, and their absence during task engagement, can reveal underlying shared functional organization. CHO can be applied to a wide range of neuroimaging techniques such as EEG, MEG, ECoG, and SEEG to elucidate the involvement of different brain regions in various cognitive functions. For example, using CHO, our study found focal specific alpha oscillations over occipital (visual) cortex in EEG and focal beta oscillations over parietal (motor) cortex

in ECoG. These results demonstrate the utility of CHO in precisely mapping the spatial distribution of neural oscillations across the brain, and in revealing shared functional organization of brain networks.

### Fundamental frequency – the 'what'

CHO revealed the fundamental frequencies of asymmetric neural oscillations recorded from the scalp, auditory cortex, motor cortex, Broca's area, and hippocampus. Distinct brain states can be identified based on the fundamental frequency of their underlying neural oscillation. CHO showed high specificity in determining the fundamental frequency of synthetic non-sinusoidal oscillations (see *Figure 4B*). When applied to ECoG and SEEG signals, CHO revealed distinct fundamental frequencies of oscillations found within auditory cortex, motor cortex, Broca's area, and hippocampus (see *Figure 9*). CHO can be applied in real time to detect the fundamental frequency and the onset/offset of neural oscillations. Characterizing neural oscillations in real time can make transitions in brain states observable to the investigator. For example, investigators can characterize brain dynamics during wakefulness, sleep, or specific cognitive tasks by tracking changes in oscillatory activity during different behavioral states. This information provides insights into the brain's adaptability and flexibility in response to internal and external cues and could inform closed-loop neuromodulation.

### Underlying mechanisms – the 'how'

Accurate detection of neural oscillations aids in deciphering the underlying mechanisms governing their generation and synchronization. In our study, we focused on determining the temporal dynamics, spatial distribution, and fundamental frequency of neural oscillations. The results of our study, and more specifically the CHO method itself, provide a methodological foundation to systematically study oscillatory connectivity and traveling oscillations throughout cortical layers and brain regions to create insights into unraveling the generating mechanism of neural oscillations. The information gained from such studies could create a better understanding of neural circuitry at the network level and could inform computational models that help refine our knowledge of the complex mechanisms underlying brain function.

### Functional purpose – the 'why'

Neural oscillation detection plays a crucial role in uncovering the functional significance of oscillatory activity. In our study, CHO detected focal alpha oscillations over occipital (visual) cortex in EEG and focal beta oscillations over frontal (motor) cortex in ECoG during the pre-stimulus period of an auditory reaction-time task (see *Figure 5* and *Figure 6*). The presence of these oscillations during the pre-stimulus period implicates visual alpha and motor beta oscillations in inhibition. We found the same inhibitory oscillatory phenomenon over the auditory cortex, however, with a fundamental frequency of 7 Hz, indicating functional independence between inhibitory oscillations found in visual, motor, and auditory cortex (see *Figure 7C and D*). The approach presented in this study could be expanded to studying attention, memory, decision-making, and more by correlating neural oscillations with specific cognitive processes. Further, applying cross-frequency and phase-amplitude coupling analysis to oscillations detected by CHO could illuminate the role of neural oscillations in facilitating information processing and communication between brain regions.

### Pathologies – the 'whom'

Detecting and characterizing neural oscillations has significant implications for the study of neurological and psychiatric disorders. For example, recent studies reported that patients affected by severe Parkinson's disease exhibited more asymmetry between peak and trough amplitudes in beta oscillations (*Cole et al., 2017*; *Jackson et al., 2019*). The high specificity demonstrated by CHO in detecting asymmetric neural oscillations could benefit the investigation of neural pathologies. Specifically, CHO could improve the quality of asymmetry measurements by providing onset/offset detection of the beta oscillations with high specificity. Abnormalities in neural oscillations are often associated with various pathologies. Detecting and characterizing aberrant oscillatory patterns could lead to identifying biomarkers for specific disorders and insights into their underlying mechanisms. These advancements could aid the development of targeted therapies and treatments for these conditions.

## Illuminating neural oscillations

Overall, developing a reliable neural oscillation detection method is crucial for advancing our understanding of brain function and cognition. The presented CHO method opens up new avenues of research by contributing to the investigation of temporal dynamics, spatial distribution, brain states, underlying mechanisms, functional purpose, and pathologies of neural oscillations. Ultimately, a comprehensive understanding of neural oscillations will deepen our knowledge of the brain's complexity and pave the way for innovative approaches to treating neurological and psychiatric disorders.

### Limitations

The results of this study show that our CHO method favors specificity over sensitivity when SNR is low. More specifically, CHO exhibited a low sensitivity due to the high false-negative rate in a low-SNR environment. This means that even though there are oscillations present in the recorded signals, CHO cannot detect them when they are drowned in noise. To investigate whether this is an issue in real-world applications, we determined the averaged SNR of alpha oscillations in EEG (–7 dB) and ECoG (–6 dB). Based on our evaluation of synthetic data, we found that at these physiologically motivated SNR levels, CHO can detect 50–60% of all true oscillations. This sensitivity could be further improved by averaging across spatially correlated locations, e.g., within the hippocampus.

One potential approach to reducing the dependency of sensitivity on SNR is to apply a wavelet transform in the estimation of the time-frequency map of the signal. Wavelet transform can better capture short cycles of oscillations. Currently, CHO uses a Hilbert transform method rather than wavelet or short-time fast Fourier transform because it is easy to implement in MATLAB and provides better control over the spectral shape (i.e. better accuracy in detecting peak frequency of oscillations, *Cohen, 2014*). Despite the theoretical advantages of wavelet over Hilbert transform, in developing our CHO method, we found no significant differences when we used different approaches to estimate the time-frequency map. This finding is further supported by a comparative study shown by *Bruns, 2004*. However, because our CHO method is modular, the FFT-based time-frequency analysis can be replaced with more sophisticated time-frequency estimation methods to improve the sensitivity of neural oscillation detection. Specifically, a state-space model (*Matsuda and Komaki, 2017*; *Beck et al., 2022*; *Brady and Bardouille, 2022*; *He et al., 2023*) or empirical mode decomposition (*Fabus et al., 2022*; *Quinn et al., 2021*) may improve the estimation of the autocorrelation of the harmonic structure underlying non-sinusoidal oscillations. Furthermore, a Gabor transform or matching pursuit-based approach may improve the onset/offset detection of short burst-like neural oscillations (*Kuś et al., 2013*; *Morales and Bowers, 2022*).

Another avenue to improve the sensitivity of CHO is to modify the third criterion to better distinguish neural oscillations from background noise. When we performed each detection step within CHO, as shown in *Figure 3*, we captured oscillations in a low-SNR situation. However, applying the third criterion rejected many possible bounding boxes. Thus, developing a better conceptual framework to reject harmonic peaks in the spectral domain may decrease the false-negative rate and, in turn, increase the sensitivity in low-SNR situations.

Another limitation of this study is that it does not assess the harmonic structure of neural oscillations. Thus, CHO cannot distinguish between oscillations that have the same fundamental frequency but differ in their non-sinusoidal properties. This limitation stems from the objective of this study, which is to identify the fundamental frequency of non-sinusoidal neural oscillations. Overcoming this limitation requires further studies to improve CHO to distinguish between different non-sinusoidal properties of pathological neural oscillations. The data that is necessary for these further studies could be obtained from the wide range of studies that have linked the harmonic structures in the neural oscillations to various cognitive functions (*van Dijk et al., 2010*; *Schalk, 2015*; *Mazaheri and Jensen, 2008*) and neural disorders (*Cole et al., 2017*; *Jackson et al., 2019*; *Hu et al., 2023*). For example, *Cole and Voytek, 2019*, showed that a harmonic structure of beta oscillations can explain the degree of Parkinson's disease, and *Hu et al., 2023*, showed the number of harmonic peaks can localize the seizure onset zone.

### Conclusions

Neural oscillations are thought to play an important role in coordinating neural activity across different brain regions, allowing for the integration of sensory information, the control of motor movements,

and the maintenance of cognitive functions. Thus, better methods to detect and characterize neural oscillations, especially those that are asymmetric, can greatly impact neuroscience. In this study, we present CHO as a method to reveal the 'when', the 'where', and the 'what' of neural oscillations. With this method, we overcome the confounding effect of detecting spurious oscillations that result from harmonics of the non-sinusoidal neural oscillations (*Donoghue et al., 2022*). In our study, we demonstrate that solving this problem yields scientific insights into local beta oscillations in pre-motor areas, the onset/offset of oscillations in the time domain, and the fundamental frequency of hippocampal oscillations. These results demonstrate the potential for CHO to support closed-loop neuromodulation (brain-computer interfaces and neurofeedback) and neural oscillation detection systems to implement various neurological diagnostic and therapeutic systems and methods.

## Materials and methods
### Electrophysiological data
Eight human subjects implanted with ECoG electrodes (x1–x8, 4 females, average age = 41±14) participated in an auditory reaction-time task at the Albany Medical Center in Albany, New York. The subjects were mentally and physically capable of participating in our study (average IQ = 96±18, range 75–120, *Wechsler, 1997*). All subjects were patients with intractable epilepsy who underwent temporary placement of subdural electrode arrays to localize seizure foci before surgical resection.

The implanted electrode grids were approved for human use (Ad-Tech Medical Corp., Racine, WI, USA; and PMT Corp., Chanhassen, MN, USA). The platinum-iridium electrodes were 4 mm in diameter (2.3 mm exposed), spaced 10 mm center-to-center, and embedded in silicone. The electrode grids were implanted in the left hemisphere for seven subjects (x1, x3, x6, and x7) and the right hemisphere for five subjects (x2, x4, x5, and x8). Following the placement of the subdural grids, each subject had postoperative anterior-posterior and lateral radiographs and computer tomography (CT) scans to verify grid location. These CT images, in conjunction with magnetic resonance imaging (MRI), were used to construct three-dimensional subject-specific cortical models and derive the electrode locations (*Coon et al., 2016*).

A further seven healthy human subjects (y1–y7, all males, average age = 27±3.6) served as a control group for which we recorded EEG while performing the same auditory reaction-time task. These subjects were fitted with an elastic cap (Electro-Cap International, *Blom and Anneveldt, 1982*) with tin (*Polich and Lawson, 1985*) scalp electrodes in 64 positions according to the modified 10–20 system (*Acharya et al., 2016*).

In addition, six human subjects implanted with ECoG electrodes (ze1–ze6, 1 female, mean age 46, range between 31 and 69) participated in resting-state recording at the Albany Medical Center in Albany, New York. All six subjects had extensive electrode coverage over the lateral STG. Patients provided informed consent to participate in the study, and additional verbal consent was given prior to each testing session. The Institutional Review Board at Albany Medical Center approved the experimental protocol. Electrodes were comprised of platinum-iridium and spaced 3–10 mm (PMT Corp., Chanhassen, MN, USA).

All ECoG and EEG subjects provided informed consent for participating in the study, which was approved by the Institutional Review Board of Albany Medical College and the Human Research Protections Office of the U.S. Army Medical Research and Materiel Command.

Lastly, six human subjects implanted with SEEG electrodes (zs1–zs6, 3 females, average age = 46±16.6) participated in resting-state recordings at the Barnes Jewish Hospital in St. Louis, MO. All subjects were patients with intractable epilepsy who underwent temporary placement of subdural electrodes to localize seizure foci prior to surgical resection. All SEEG subjects provided informed consent for participating in the study, which was approved by the Institutional Review Board of Washington University School of Medicine in St. Louis.

The implanted SEEG electrodes were approved for human use (Ad-Tech Medical Corp., Racine, WI, USA; and PMT Corp., Chanhassen, MN, USA). The platinum-iridium electrodes were 2 mm in length (0.8 mm diameter) and spaced 3.5–5 mm center-to-center. Following the placement of the SEEG electrodes, each subject had postoperative anterior-posterior and lateral radiographs and CT scans to verify electrode locations. These postoperative CT images, in conjunction with preoperative MRI,

were used to construct three-dimensional subject-specific cortical models and derive the electrode locations (*Coon et al., 2016*).

## Data collection

We recorded EEG, ECoG, and SEEG signals from the subjects at their bedside using the general-purpose Brain-Computer Interface (BCI2000) software (*Schalk et al., 2004*), interfaced with eight 16-channel g.USBamp biosignal acquisition devices (for EEG), one 256-channel g.HIamp biosignal acquisition device (g.tec., Graz, Austria, for ECoG), or one Nihon Kohden JE-120A long-term recording system (Nihon Kohden, Tokyo, Japan, for SEEG) to amplify, digitize (sampling rate 1200 Hz for EEG and ECoG and 2000 Hz for SEEG), and store the signals. To ensure safe clinical monitoring of ECoG signals during the experimental tasks, a connector split the cables connected to the patients into a subset connected to the clinical monitoring system and a subset connected to the amplifiers.

## Task

The subjects performed an auditory reaction task, responding with a button press to a salient 1 kHz tone. For their response, the subjects used their thumb contralateral to their ECoG implant. In total, the subjects performed between 134 and 580 trials. Throughout each trial, the subjects were first required to fixate and gaze at the screen in front of them. Next, a visual cue indicated the trial's start, followed by a random 1–3 s pre-stimulus interval and, subsequently, the auditory stimulus. The stimulus was terminated by the subject's button press or after a 2 s time-out, after which the subject received feedback about his/her reaction time. This feedback motivated the subjects to respond as quickly as possible to the stimulus. We penalized subjects with a warning tone to prevent false starts if they responded too fast (i.e. less than 100 ms after stimulus onset). We excluded false-start trials from our analysis. We were interested in this task's auditory and motor responses in this study. This required defining the onset of these two responses. We time-locked our analysis of the auditory response to the onset of the auditory stimulus (as measured by the voltage between the sound port on the PC and the loudspeaker). For the motor response, we time-locked our analysis to the time when the push button was pressed. To ensure the temporal accuracy of these two onset markers, we sampled them simultaneously with the EEG/ECoG signals using dedicated inputs in our biosignal acquisition systems. We defined baseline and task periods for the auditory and motor response. Specifically, we used the 0.5 s period prior to the stimulus onset as the baseline for the auditory response and the 1 to 0.5 s period prior to the button press as the baseline for the motor response. Similarly, we used the 1 s period after stimulus onset as the task period for the auditory response and the period from 0.5 s before to 0.5 s after the button press as the task period for the motor task.

## Data pre-processing

As our amplifiers acquired raw, unfiltered EEG/ECoG/SEEG signals, we removed any offset from our signals using a second-order Butterworth high-pass filter at 0.05 Hz. Next, we removed any common noise using a common median reference filter (*Liu et al., 2015*). To create the common-mode reference, we excluded signals that exhibited an excessive 60 Hz line noise level (i.e. 10 times the median absolute deviation). To improve the SNR of our recordings and to reduce the computational complexity of our subsequent analysis, we downsampled our signals from 1200 or 2000 Hz to 400 or 500 Hz, respectively, using MATLAB's 'resample' function, which uses a polyphase antialiasing filter to resample the signal at the uniform sample rate.

## Phase-phase coupling

To demonstrate phase-locking, as illustrated between theta and beta oscillations in *Figure 1E* and *Figure 1K*, we utilized the *n:m* phase-phase coupling method described in *Belluscio et al., 2012*. Specifically, we calculated the 'mean radial distance': $R_{n:m} = \|\frac{1}{N}\sum_{j=1}^{N} e^{i\Delta\phi_{nm}(t_j)}\|$, where $j$ indexes the samples in time, and $N$ represents the number of samples (epoch length in seconds × sampling frequency in Hz). $R_{n:m}$ equals 1 when $\Delta\phi_{nm}(t_j)$ is constant for all time samples $t_j$, and 0 when $\Delta\phi_{nm}$ is uniformly distributed. Of note, $\Delta\phi_{nm}(t_j)$ equals $n\phi_{f_1}(t_j) - m\phi_{f_2}(t_j)$, with $f_1$ and $f_2$ being two different frequency bands.

## A novel oscillation detection method

We propose a novel method based on principle criteria to identify the 'when', 'where', and 'what' of neural oscillations. The principle criteria are as follows: (1) Oscillations (peaks over 1/*f* noise) must

be present in the time and frequency domain. (2) Oscillations must exhibit at least 2 full cycles. (3) The periodicity of an oscillation is the fundamental frequency of the oscillation. The procedural steps of CHO adhere to these principle criteria, as shown in *Figure 3* and *Algorithm* 1. First, we apply a time-frequency analysis to determine power changes for each frequency component over time. To measure the significant spectral power increase over the time domain, we use the $1/f$ fitting technique as the principal threshold. In other words, the proposed method only considers those oscillations that emerge above the underlying $1/f$ noise. Thus, any oscillation with smaller power than $1/f$ noise is not considered to be an oscillation. To accomplish this, we subtract the underlying $1/f$ noise within the time-frequency domain. Specifically, we divide the time domain into four periods and estimate the minimum $1/f$ aperiodic fit across these periods (see Line 5 in *Algorithm* 1). After the subtraction of the underlying $1/f$ noise, we calculate the averaged power difference between the signal and the $1/f$ noise (named sigma). If the spectral power exceeds two times sigma, we consider the oscillation to exhibit significant power above the $1/f$ noise (see Line 12 in *Algorithm* 1). Next, we cluster time points with significant power over $1/f$ noise to generate initial bounding boxes as shown in *Figure 3A*; this idea is adopted from a previous study (*Neymotin et al., 2022*) (see Lines 10–20 in *Algorithm* 1).

---

**Algorithm 1.** CHO detection method

---

1: **procedure** CHO
2: Let $x(t)$ denote a signal at time point $t \in T$.
3: *Remove 1/f noise in time-frequency map:*
4: $P(t,f) \leftarrow$ log power of $x(t)$ at frequency $f \in F$
5: $T_1, ..., T_N \in T \leftarrow$ segment $T$ into $N$ windows
6: $b_i, e_i \leftarrow$ offset $b$ and exponent $e$ of $1/f$ fitting from $P(T_{i \in T}, F)$
7: $t_{min} \leftarrow argmin_i b_i$
8: $L_{min} \leftarrow b(t_{min}) - log(F^{e_{min}})$
9: $P'(t,F) \leftarrow P(t,F) - L_{min}$
10: *Generate initial bounding boxes:*
11: $\sigma(f) \leftarrow$ standard deviation of $P'(t,f)$ over $t$
12: $C_{k \in K} \leftarrow$ cluster the data points in $P'(t,f)$ if $P'(t,f) > 2\sigma(f)$
13: $B_{k \in K} \leftarrow$ generate $K$ initial bounding boxes
14: $B_k.cf \leftarrow$ center frequency of the bounding box
15: $B_k.ct \leftarrow$ center time point of the bounding box
16: $B_k.power \leftarrow$ peak power within the bounding box
17: $B_k.minf \leftarrow$ lower bound frequency of the bounding box
18: $B_k.maxf \leftarrow$ upper bound frequency of the bounding box
19: $B_k.start \leftarrow$ onset time of the bounding box
20: $B_k.stop \leftarrow$ offset time of the bounding box
21: *Reject boxes have short cycles:*
22: $Cycle_{k \in K} \leftarrow B_k.cf \times (B_k.stop - B_k.start)$
23: $B_{m \in M} \leftarrow$ reject $B_k$ if $Cycle_k < 2$
24: *Reject boxes if its periodicity of raw signal and center frequency are different:*
25: $A_m(l) \leftarrow$ auto-correlation of the raw signal $x(t')$, $t' \in T'$, where $T' = B_m.start, ..., B_m.stop$
26: $Ppeaks_m, Npeaks_m \leftarrow$ Sets of positive and negative peaks in $A_m(l)$
27: $Pinterval_m, Ninterval_m \leftarrow$ Intervals of $Peaks_m$ and $Npeaks_m$, respectively
28: $Periodicity_m \leftarrow$ Periodicity (Hz) of $Pinterval_m$
29: $Psimilarity_m, Nsimilarity_m \leftarrow$ Similarity (%) of $Pinterval_m$ and $Ninterval_m$, respectively
30: $B_{h \in H} \leftarrow$ Accept $B_m$ if $B_m.minf < Periodicity_m < B_m.maxf$ and $Psimilarity_m < 30\%$
31: $B_{j \in J} \leftarrow$ merge remained boxes if t' overlaps > 75% each other
32: Return $B_{j \in J}$

---

Next, as the second principle criterion, we only consider those oscillations that exhibit at least 2 full cycles. This restriction allows CHO to distinguish oscillations from confounding ERPs or EPs. In general, the frequency characteristics of those potentials often overlap with neural oscillations (e.g. theta power of ERPs and theta power of theta rhythm). However, ERPs or EPs never exhibit more than 2 cycles (see Line 23 in *Algorithm* 1). Therefore, we reject those bounding boxes that exhibit less than 2 cycles. An example is shown in *Figure 3B*.

Lastly, we calculate the periodicity of an oscillation using an autocorrelation analysis to determine the fundamental frequency of the oscillation. Non-sinusoidal signals are known to exhibit harmonics in the frequency domain, significantly increasing the false-positive detection rate – the confounding factor addressed by CHO's third criterion. The power spectrum of the non-sinusoidal oscillations has additional harmonic peaks over $1/f$ noise, even though the periodicity of the signal does not match the harmonic peak frequency. Therefore, the positive peaks of the oscillation's autocorrelation represent the oscillation's periodicity and fundamental frequency (see *Figure 2*). As shown in *Figure 3C*,

the center frequency of the bounding box is 24 Hz, but the periodicity of the raw signal within the bounding box does not match 24 Hz. Consequently, this bounding box will be rejected (see Line 30 in *Algorithm* 1). Finally, the method merges those remaining bounding boxes that neighbor each other in the frequency domain and that overlap more than 75% in time (*Neymotin et al., 2022*).

The MATLAB code that implements CHO and sample data is available on GitHub (https://github.com/neurotechcenter/CHO, copy archived at *Cho, 2023*).

## Trade-offs in adjusting the hyper-parameters that govern the detection in CHO

The ability of CHO to detect neural oscillations and determine their fundamental frequency is governed by four principal hyper-parameters. Adjusting these parameters requires understanding their effect on the sensitivity and specificity in the detection of neural oscillations.

The first hyper-parameter is the number of time windows (*N* in Line 5 in *Algorithm* 1), that is used to estimate the 1/*f* noise. In our performance assessment of CHO, we used four windows, resulting in estimation periods of 250 ms in duration for each 1/*f* spectrum. A higher number of time windows results in smaller estimation periods and thus minimizes the likelihood of observing multiple neural oscillations within this time window, which otherwise could confound the 1/*f* estimation. However, a higher number of time windows and, thus, smaller time estimation periods may lead to unstable 1/*f* estimates.

The second hyper-parameter defines the minimum number of cycles of a neural oscillation to be detected by CHO (see Line 23 in *Algorithm* 1). In our study, we specified this parameter to be 2 cycles. Increasing the number of cycles increases specificity, as it will reject spurious oscillations. However, increasing the number also reduces sensitivity as it will reject short oscillations.

The third hyper-parameter is the significance threshold that selects positive peaks within the autocorrelation of the signal. The magnitude of the peaks in the autocorrelation indicates the periodicity of the oscillations (see Line 26 in *Algorithm* 1). Referred to as 'NumSTD', this parameter denotes the number of standard deviations that a positive peak has to exceed to be selected to be a true oscillation. For this study, we set the 'NumSTD' value to 1. Increasing the 'NumSTD' value increases specificity in the detection as it reduces the detection of spurious peaks in the autocorrelation. However, increasing the 'NumSTD' value also decreases the sensitivity in the detection of neural oscillations with varying instantaneous oscillatory frequencies.

The fourth hyper-parameter is the percentage of overlap between two bounding boxes that trigger their merger (see Line 31 in *Algorithm* 1). In our study, we set this parameter to 75% overlap. Increasing this threshold yields more fragmentation in the detection of oscillations, while decreasing this threshold may reduce the accuracy in determining the onset and offset of neural oscillations.

## Validation on synthetic non-sinusoidal oscillations

While empirical physiological signals are most appropriate for validating our method, they generally lack the necessary ground truth to characterize neural oscillation with sinusoidal or non-sinusoidal properties. To overcome this limitation, we first validated CHO on synthetic non-sinusoidal oscillatory bursts (2.5 cycles, 1–3 s long) convolved with 1/*f* noise to test the performance of the proposed method.

As shown in *Figure 4*, we generated 5-s-long periods comprised of 1/*f* noise (i.e. pink noise). We added non-sinusoidal oscillations with different amplitudes and lengths. The amplitudes of non-sinusoidal oscillations vary between 5 and 20 µV, while the pink noise remains at 10 µV in amplitude. The SNR was calculated by the snr() function in the Signal Processing Toolbox of MATLAB, which determines the SNR in decibels of the non-sinusoidal burst by computing the ratio between summed squared magnitudes of the oscillation and the pink noise, respectively. We simulated 10 iterations for each amplitude. For each iteration, we tested four different lengths of non-sinusoidal oscillations (1 cycle, 2.5 cycles, 1 s, and 3 s long).

We generated non-sinusoidal oscillations by introducing asymmetry between the trough and peak periods of sinusoidal waves. To generate this asymmetric nature of an oscillation, we applied a 9:1 ratio between trough and peak amplitudes, as shown in an example of *Figure 4A*. To smooth the onset and offset of the non-sinusoidal oscillations, we used Tukey (tapered cosine) window function

with a 0.40 ratio for the taper section (*Bloomfield, 2004*). Of note, the smaller the Tukey ratio within the taper section, the higher the occurrence of high-frequency artifacts.

To evaluate the performance of CHO, we calculated the specificity and sensitivity of CHO in detecting non-sinusoidal oscillations. High specificity depends on high true-negative and low false-positive detection rates. In contrast, high sensitivity depends on high true-positive and low false-negative detection rates. In this simulation, we expected harmonic oscillations to increase the false-positive detection rate, and one-cycled oscillations to decrease the true-negative detection rate within conventional methods. Thus, harmonic oscillations and one-cycled oscillations decrease the specificity, not sensitivity.

For evaluating the performance of each method in determining the fundamental frequency of the oscillations, we defined an accurate detection as one that exhibited a difference between the ground truth peak frequency and detected frequency of less than 1.5 Hz. Furthermore, to evaluate the performance of each method in detecting the onset/offset of the oscillations, we calculated the correlation between the envelope of the ground truth oscillation and the detected oscillation. We defined those onset/offset detections as accurate if the correlation was positive and the p-value was smaller than 0.05.

## Code availability

The MATLAB code and sample data used for CHO are available at https://github.com/neurotech-center/CHO, (copy archived at *Cho, 2023*).

## Acknowledgements

This work was supported by the National Institutes of Health (NIH) grants R01-MH120194, R01-EB026439, U24-NS109103, U01-NS108916, U01-NS128612, P41-EB018783, the McDonnell Center for Systems Neuroscience and Fondazione Neurone.

## Additional information

### Competing interests

Hohyun Cho: One U.S. patent (Provisional Application Serial No.63/326,257) related to systems and methodsfor detection of neurophysiological signal oscillations described in this manuscript was filed on March 31, 2022. The inventors/contributors of this patent involve some of themanuscript authors, including HC, MA, JTW, PB. Peter Brunner: One U.S. patent (Provisional Application Serial No.63/326,257) related to systems and methods. The other authors declare that no competing interests exist.

### Funding

| Funder | Grant reference number | Author |
|---|---|---|
| National Institutes of Health | R01-MH120194 | Jon T Willie |
| National Institutes of Health | R01-EB026439 | Peter Brunner |
| National Institutes of Health | U24-NS109103 | Markus Adamek<br>Peter Brunner |
| National Institutes of Health | U01-NS108916 | Hohyun Cho<br>Markus Adamek<br>Peter Brunner |
| National Institutes of Health | U01-NS128612 | Peter Brunner |
| National Institutes of Health | P41-EB018783 | Hohyun Cho<br>Markus Adamek<br>Jon T Willie<br>Peter Brunner |

| Funder | Grant reference number | Author |
|---|---|---|
| McDonnell Center for Systems Neuroscience | | Hohyun Cho Peter Brunner |
| Fondazione Neurone | | Peter Brunner |

The funders had no role in study design, data collection and interpretation, or the decision to submit the work for publication.

## Author contributions

Hohyun Cho, Conceptualization, Formal analysis, Visualization, Methodology, Writing – original draft, Writing – review and editing; Markus Adamek, Visualization, Methodology, Writing – review and editing; Jon T Willie, Funding acquisition, Methodology, Writing – review and editing; Peter Brunner, Conceptualization, Data curation, Funding acquisition, Methodology, Writing – original draft, Writing – review and editing

## Author ORCIDs

Hohyun Cho (ID) https://orcid.org/0000-0003-1512-0622
Markus Adamek (ID) https://orcid.org/0000-0001-8519-9212
Peter Brunner (ID) https://orcid.org/0000-0002-2588-2754

## Ethics

All ECoG and EEG subjects provided informed consent for participating in the study, which was approved by the Institutional Review Board of Albany Medical College and the Human Research Protections Office of the U.S. Army Medical Research and Materiel Command. All SEEG subjects provided informed consent for participating in the study, which was approved by the Institutional Review Board of Washington University School of Medicine in St. Louis.

Reviewer #1 (Public Review): https://doi.org/10.7554/eLife.91605.3.sa1
Reviewer #2 (Public Review): https://doi.org/10.7554/eLife.91605.3.sa2
Author response https://doi.org/10.7554/eLife.91605.3.sa3

## Additional files

### Supplementary files
• MDAR checklist

### Data availability

The Matlab code and sample data used for CHO are available at https://github.com/neurotechcenter/CHO (copy archived at *Cho, 2023*). The ECoG dataset accompanying this manuscript has been deposited in an online repository (https://doi.org/10.5281/zenodo.4361654).

The following dataset was generated:

| Author(s) | Year | Dataset title | Dataset URL | Database and Identifier |
|---|---|---|---|---|
| Cho H, Schalk G, Adamek M, Moheimanian L, Coon WG, Jun SC, Wolpaw JR, Brunner P | 2020 | Dataset and code accompanying publication: "Revealing the Physiological Origin of Event-Related Potentials using Electrocorticography in Humans" | https://doi.org/10.5281/zenodo.4361654 | Zenodo, 10.5281/zenodo.4361654 |

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
