## [Editor Report · eLife assessment]

Building on previous toolboxes to distinguish 1/f noise from oscillatory activity, this study introduces an **important** advancement in neural signal analysis to identify oscillatory activity in electrophysiological data that refines the accuracy of identifying non-sinusoidal neural oscillations. Extensive validation, using synthetic and various empirical data, provides **convincing** evidence for the accuracy of the method and outlines practical implications for relevant scientific problems in the field.

---

## [Referee Report · Reviewer #1 (Public Review)]

Summary:

The study introduces and validates the Cyclic Homogeneous Oscillation (CHO) detection method to precisely determine the duration, location, and fundamental frequency of non-sinusoidal neural oscillations. Traditional spectral analysis methods face challenges in distinguishing the fundamental frequency of non-sinusoidal oscillations from their harmonics, leading to potential inaccuracies. The authors implement an underexplored approach, using the auto-correlation structure to identify the characteristic frequency of an oscillation. By combining this strategy with existing time-frequency tools to identify when oscillations occur, the authors strive to solve outstanding challenges involving spurious harmonic peaks detected in time-frequency representations. Empirical tests using electrocorticographic (ECoG) and electroencephalographic (EEG) signals further support the efficacy of CHO in detecting neural oscillations.

Strengths:

The paper puts important emphasis on the 'identity' question of oscillatory identification. The field primarily identifies oscillations through frequency, space (brain region), and time (length, and relative to task or rest). However, more tools that claim to further characterize oscillations by their defining/identifying traits are needed, in addition to data-driven studies about what the identifiable traits of neural oscillations are beyond frequency, location, and time. Such tools are useful for potentially distinguishing between circuit mechanistic generators underlying signals that may not otherwise be distinguished. This paper states this problem well and puts forth a new type of objective for neural signal processing methods.

The paper uses synthetic data and multimodal recordings at multiple scales to validate the tool, suggesting CHO's robustness and applicability in various real-data scenarios. The figures illustratively demonstrate how CHO works on such synthetic and real examples, depicting in both time and frequency domains. The synthetic data are well-designed, and capable of producing transient oscillatory bursts with non-sinusoidal characteristics within 1/f noise. Using both non-invasive and invasive signals exposes CHO to conditions which may differ in the extent and quality of harmonic signal structure. An interesting follow-up question is whether the utility demonstrated here holds for MEG signals, as well as source-reconstructed signals from non-invasive recordings.

This study is accompanied by open-source code and data for use by the community.

Weaknesses:

The criteria that the authors use for neural oscillations embody some operating assumptions underlying their characteristics, perhaps informed by immediate use cases intended by the authors (e.g., hippocampal bursts). The extent to which these assumptions hold in all circumstances should be investigated. For instance, the notion of consistent auto-correlation breaks down in scenarios where instantaneous frequency fluctuates significantly at the scale of a few cycles. Imagine an alpha-beta complex without harmonics (Jones 2009). If oscillations change phase position within a timeframe of a few cycles, it would be difficult for a single peak in the auto-correlation structure to elucidate the complex time-varying peak frequency in a dynamic fashion. Likewise, it is unclear whether bounding boxes with a pre-specified overlap can capture complexes that manoeuvre across peak frequencies.

This method appears to lack the implementation of statistical inferential techniques for estimating and interpreting auto-correlation and spectral structure. In standard practice, auto-correlation functions and spectral measures can be subjected to statistical inference to establish confidence intervals, often helping to determine the significance of the estimates. Doing so would be useful for expressing the likelihood that an oscillation and its harmonic has the same auto-correlation structure and fundamental frequency, or more robustly identifying harmonic peaks in the presence of spectral noise. Here, the authors appear to use auto-correlation and time-frequency decomposition more as a deterministic tool rather than an inferential one. Overall, an inferential approach would help differentiate between true effects and those that might spuriously occur due to the nature of the data. Ultimately, a more statistically principled approach might estimate harmonic structure in the presence of noise in a unified manner transmitted throughout the methodological steps.

---

## [Referee Report · Reviewer #2 (Public Review)]

Summary:

A new toolbox is presented that builds on previous toolboxes to distinguish between real and spurious oscillatory activity, which can be induced by non-sinusoidal waveshapes. Whilst there are many toolboxes that help to distinguish between 1/f noise and oscillations, not many tools are available that help to distinguish true oscillatory activity from spurious oscillatory activity induced in harmonics of the fundamental frequency by non-sinusoidal waveshapes. The authors present a new algorithm which is based on autocorrelation to separate real from spurious oscillatory activity. The algorithm is extensively validated using synthetic (simulated) data, and various empirical datasets from EEG, and intracranial EEG in various locations and domains (i.e. auditory cortex, hippocampus, etc.).

Strengths:

Distinguishing real from spurious oscillatory activity due to non-sinusoidal waveshapes is an issue that has plagued the field for quite a long time. The presented toolbox addresses this fundamental problem which will be of great use for the community. The paper is written in a very accessible and clear way so that readers less familiar with the intricacies of Fourier transform and signal processing will also be able to follow it. A particular strength is the broad validation of the toolbox, using synthetic, scalp EEG, EcoG, and stereotactic EEG in various locations and paradigms.

Weaknesses:

A weakness is that the algorithm seems to be quite conservative in identifying oscillatory activity which may render it only useful for analyzing very strong oscillatory signals (i.e. alpha), but less suitable for weaker oscillatory signals (i.e. gamma).

---

## [Author Response]

The following is the authors’ response to the original reviews.

**Public Reviews:**

**Reviewer #1 (Public Review):**
Summary:The study introduces and validates the Cyclic Homogeneous Oscillation (CHO) detection method to precisely determine the duration, location, and fundamental frequency of non-sinusoidal neural oscillations. Traditional spectral analysis methods face challenges in distinguishing the fundamental frequency of non-sinusoidal oscillations from their harmonics, leading to potential inaccuracies. The authors implement an underexplored approach, using the auto-correlation structure to identify the characteristic frequency of an oscillation. By combining this strategy with existing time-frequency tools to identify when oscillations occur, the authors strive to solve outstanding challenges involving spurious harmonic peaks detected in time-frequency representations. Empirical tests using electrocorticographic (ECoG) and electroencephalographic (EEG) signals further support the efficacy of CHO in detecting neural oscillations.

Response: We thank the reviewer for recognizing the strengths of our method in this encouraging review and for the opportunity to further improve and finalize our manuscript.

Strengths:(1) The paper puts an important emphasis on the 'identity' question of oscillatory identification. The field primarily identifies oscillations through frequency, space (brain region), and time (length, and relative to task or rest). However, more tools that claim to further characterize oscillations by their defining/identifying traits are needed, in addition to data-driven studies about what the identifiable traits of neural oscillations are beyond frequency, location, and time. Such tools are useful for potentially distinguishing between circuit mechanistic generators underlying signals that may not otherwise be distinguished. This paper states this problem well and puts forth a new type of objective for neural signal processing methods.

Response: We sincerely appreciate this encouraging summary of the objective of our manuscript.

(2) The paper uses synthetic data and multimodal recordings at multiple scales to validate the tool, suggesting CHO's robustness and applicability in various real-data scenarios. The figures illustratively demonstrate how CHO works on such synthetic and real examples, depicting in both time and frequency domains. The synthetic data are well-designed, and capable of producing transient oscillatory bursts with non-sinusoidal characteristics within 1/f noise. Using both non-invasive and invasive signals exposes CHO to conditions which may differ in extent and quality of the harmonic signal structure. An interesting followup question is whether the utility demonstrated here holds for MEG signals, as well as source-reconstructed signals from non-invasive recordings.

Response: We thank the reviewer for this excellent suggestion. Indeed, our next paper will focus on applying our CHO method to signals that were source-reconstructed from non-invasive recordings (e.g., MEG and EEG) to extract their periodic activity.

(3) This study is accompanied by open-source code and data for use by the community.

Response: We thank the reviewer for recognizing our effort to widely disseminate our method to the broader community.

Weaknesses:(1) Due to the proliferation of neural signal processing techniques that have been designed to tackle issues such as harmonic activity, transient and event-like oscillations, and non-sinusoidal waveforms, it is naturally difficult for every introduction of a new tool to include exhaustive comparisons of all others. Here, some additional comparisons may be considered for the sake of context, a selection of which follows, biased by the previous exposure of this reviewer. One emerging approach that may be considered is known as state-space models with oscillatory and autoregressive components (Matsuda 2017, Beck 2022). State-space models such as autoregressive models have long been used to estimate the auto-correlation structure of a signal. State-space oscillators have recently been applied to transient oscillations such as sleep spindles (He 2023). Therefore, state-space oscillators extended with auto-regressive components may be able to perform the functions of the present tool through different means by circumventing the need to identify them in time-frequency. Another tool that should be mentioned is called PAPTO (Brady 2022). Although PAPTO does not address harmonics, it detects oscillatory events in the presence of 1/f background activity. Lastly, empirical mode decomposition (EMD) approaches have been studied in the context of neural harmonics and nonsinusoidal activity (Quinn 2021, Fabus 2022). EMD has an intrinsic relationship with extrema finding, in contrast with the present technique. In summary, the existence of methods such as PAPTO shows that researchers are converging on similar approaches to tackle similar problems. The existence of time-domain approaches such as state-space oscillators and EMD indicates that the field of timeseries analysis may yield even more approaches that are conceptually distinct and may theoretically circumvent the methodology of this tool.

Response: We thank the reviewer for this valuable insight. In our manuscript, we acknowledge emerging approaches that employ state-space models or EMD for time-frequency analysis. However, it's crucial to clarify that the primary focus in our study is on the detection and identification of the fundamental frequency, as well as the onset/offset of non-sinusoidal neural oscillations. Thus, our emphasis lies specifically on these aspects. We hope that future studies will use our methods as the basis to develop better methods for time-frequency analysis that will lead to a deeper understanding of harmonic structures.

Our Limitation section is addressing this issue. Specifically, we recognize that a more sophisticated time-frequency analysis could contribute to improved sensitivity and that the core claim of our study is centered around the concept of increasing specificity in the detection of non-sinusoidal oscillations. We hope that future studies will use this as a basis for improving time-frequency analysis in general. Notably, our open-source code will greatly enable these future studies in this endeavor. Specifically, in the first step of our algorithm, the timefrequency estimation can be replaced with any other preferred time-frequency analysis, such as state-space models, EMD, Wavelet transform, Gabor transform, and Matching Pursuit.

For our own follow-up study, we plan to conduct a thorough review and comparison of emerging approaches employing state-space models or EMD for time-frequency analysis. In this study, we aim to identify which approach, including the six methods mentioned by the reviewer (Matsuda 2017, Beck 2022, He 2023, Brady 2022, Quinn 2021, and Fabus 2022), can maximize the estimation of the fundamental frequency of non-sinusoidal neural oscillations using CHO. The insights provided by the reviewer are appreciated, and we will carefully consider these aspects in our follow-up study.

In the revision of this manuscript, we are setting the stage for these future studies. Specifically, we added a discussion paragraph within the Limitation section about the state-space model, and EMD approaches:

“However, because our CHO method is modular, the FFT-based time-frequency analysis can be replaced with more sophisticated time-frequency estimation methods to improve the sensitivity of neural oscillation detection. Specifically, a state-space model (Matsuda 2017, Beck 2022, He 2023, Brady 2022) or empirical mode decomposition (EMD, Quinn 2021, Fabus 2022) may improve the estimation of the auto-correlation of the harmonic structure underlying nonsinusoidal oscillations. Furthermore, a Gabor transform or matching pursuit-based approach may improve the onset/offset detection of short burst-like neural oscillations (Kus 2013 and Morales 2022).”

(2) The criteria that the authors use for neural oscillations embody some operating assumptions underlying their characteristics, perhaps informed by immediate use cases intended by the authors (e.g., hippocampal bursts). The extent to which these assumptions hold in all circumstances should be investigated. For instance, the notion of consistent auto-correlation breaks down in scenarios where instantaneous frequency fluctuates significantly at the scale of a few cycles. Imagine an alpha-beta complex without harmonics (Jones 2009). If oscillations change phase position within a timeframe of a few cycles, it would be difficult for a single peak in the auto-correlation structure to elucidate the complex time-varying peak frequency in a dynamic fashion. Likewise, it is unclear whether bounding boxes with a pre-specified overlap can capture complexes that maneuver across peak frequencies.

Response: We thank the reviewer for this valuable insight into the methodological limitations in the detection of neural oscillations that exhibit significant fluctuations in their instantaneous frequency. Indeed, our CHO method is also limited in the ability to detect oscillations with fluctuating instantaneous frequencies. This is because CHO uses an auto-correlation-based approach to detect neural oscillations that exhibit two or more cycles. If oscillations change phase position within a timeframe of a few cycles, CHO cannot detect the oscillation because the periodicity is not expressed within the auto-correlation. This limitation can be partially overcome by relaxing the detection threshold (see Line 30 of Algorithm 1 in the revised manuscript) for the auto-correlation analysis. However, relaxing the detection threshold, in consequence, increases the probability of detecting other aperiodic activity as well. To clarify how CHO determines the periodicity of oscillations, and to educate the reader about the tradeoff between detecting oscillations with fluctuating instantaneous frequencies and avoiding detecting other aperiod activity, we have added pseudo code and a new subsection in the Methods.

A new subsection titled “Tradeoffs in adjusting the hyper-parameters that govern the detection in CHO”.

“The ability of CHO to detect neural oscillations and determine their fundamental frequency is governed by four principal hyper-parameters. Adjusting these parameters requires understanding their effect on the sensitivity and specificity in the detection of neural oscillations.

The first hyper-parameter is the number of time windows (N in Line 5 in Algorithm 1), that is used to estimate the 1/f noise. In our performance assessment of CHO, we used four windows, resulting in estimation periods of 250 ms in duration for each 1/f spectrum. A higher number of time windows results in smaller estimation periods and thus minimizes the likelihood of observing multiple neural oscillations within this time window, which otherwise could confound the 1/f estimation. However, a higher number of time windows and, thus, smaller time estimation periods may lead to unstable 1/f estimates.

The second hyper-parameter defines the minimum number of cycles of a neural oscillation to be detected by CHO (see Line 23 in Algorithm 1). In our study, we specified this parameter to be two cycles. Increasing the number of cycles increases specificity, as it will reject spurious oscillations. However, increasing the number also reduces sensitivity as it will reject short oscillations.

The third hyper-parameter is the significance threshold that selects positive peaks within the auto-correlation of the signal. The magnitude of the peaks in the auto-correlation indicates the periodicity of the oscillations (see Line 26 in Algorithm 1). Referred to as "NumSTD," this parameter denotes the number of standard errors that a positive peak has to exceed to be selected to be a true oscillation. For this study, we set the "NumSTD" value to 1. Increasing the "NumSTD" value increases specificity in the detection as it reduces the detection of spurious peaks in the auto-correlation. However, increasing the "NumSTD" value also decreases the sensitivity in the detection of neural oscillations with varying instantaneous oscillatory frequencies.

The fourth hyper-parameter is the percentage of overlap between two bounding boxes that trigger their merger (see Line 31 in Algorithm 1). In our study, we set this parameter to 75% overlap. Increasing this threshold yields more fragmentation in the detection of oscillations, while decreasing this threshold may reduce the accuracy in determining the onset and offset of neural oscillations.”

(3) Related to the last item, this method appears to lack implementation of statistical inferential techniques for estimating and interpreting auto-correlation and spectral structure. In standard practice, auto-correlation functions and spectral measures can be subjected to statistical inference to establish confidence intervals, often helping to determine the significance of the estimates. Doing so would be useful for expressing the likelihood that an oscillation and its harmonic has the same autocorrelation structure and fundamental frequency, or more robustly identifying harmonic peaks in the presence of spectral noise. Here, the authors appear to use auto-correlation and time-frequency decomposition more as a deterministic tool rather than an inferential one. Overall, an inferential approach would help differentiate between true effects and those that might spuriously occur due to the nature of the data. Ultimately, a more statistically principled approach might estimate harmonic structure in the presence of noise in a unified manner transmitted throughout the methodological steps.

Response: We thank the reviewer for sharing this insight on further enhancing our method. Indeed, CHO does not make use of statistical inferential statistics to estimate and interpret the auto-correlation and underlying spectral structure of the neural oscillation. Implementing this approach within CHO would require calculating phase-phase coupling across all cross-frequency bands and bounding boxes. However, as mentioned in the introduction section and Figure 1GL, phase-phase coupling analysis cannot fully ascertain whether the oscillations are phaselocked and thus are harmonics or, indeed, independent oscillations. This ambiguity, combined with the exorbitant computational complexity of the entailed permutation test and the requirement to perform the analysis across all cross-frequency bands, channels, and trials, makes phase-phase coupling impracticable in determining the fundamental frequency of neural oscillations in real-time and, thus, the use in closed-loop neuromodulation applications. Thus, within our study, we prioritized determining the fundamental frequency without considering the structure of harmonics.

An inferential approach can be implemented by adjusting the significance threshold that selects positive peaks within the auto-correlation of the signal. Currently, this threshold is set to represent the approximate confidence bounds of the periodicity of the fundamental frequency. To clarify this issue, we added additional pseudo code and a new subsection, titled “Tradeoffs in adjusting the hyper-parameters that govern the detection in CHO,” in the Methods section.

In future studies, we will investigate the harmonic structure of neural oscillations based on a large data set. This exploration will help us understand how non-sinusoidal properties may influence the harmonic structure. Your input is highly appreciated, and we will diligently incorporate these considerations into our research.

See Author response image 1.

**Author response image 1. sa3fig1:** Algorithm 1.

A new subsection titled “Tradeoffs in adjusting the hyper-parameters that govern the detection in CHO”.

“The ability of CHO to detect neural oscillations and determine their fundamental frequency is governed by four principal hyper-parameters. Adjusting these parameters requires understanding their effect on the sensitivity and specificity in the detection of neural oscillations.

The first hyper-parameter is the number of time windows (N in Line 5 in Algorithm 1), that is used to estimate the 1/f noise. In our performance assessment of CHO, we used four windows, resulting in estimation periods of 250 ms in duration for each 1/f spectrum. A higher number of time windows results in smaller estimation periods and thus minimizes the likelihood of observing multiple neural oscillations within this time window, which otherwise could confound the 1/f estimation. However, a higher number of time windows and, thus, smaller time estimation periods may lead to unstable 1/f estimates.

The second hyper-parameter defines the minimum number of cycles of a neural oscillation to be detected by CHO (see Line 23 in Algorithm 1). In our study, we specified this parameter to be two cycles. Increasing the number of cycles increases specificity, as it will reject spurious oscillations. However, increasing the number also reduces sensitivity as it will reject short oscillations.

The third hyper-parameter is the significance threshold that selects positive peaks within the auto-correlation of the signal. The magnitude of the peaks in the auto-correlation indicates the periodicity of the oscillations (see Line 26 in Algorithm 1). Referred to as "NumSTD," this parameter denotes the number of standard errors that a positive peak has to exceed to be selected to be a true oscillation. For this study, we set the "NumSTD" value to 1. Increasing the "NumSTD" value increases specificity in the detection as it reduces the detection of spurious peaks in the auto-correlation. However, increasing the "NumSTD" value also decreases the sensitivity in the detection of neural oscillations with varying instantaneous oscillatory frequencies.

The fourth hyper-parameter is the percentage of overlap between two bounding boxes that trigger their merger (see Line 31 in Algorithm 1). In our study, we set this parameter to 75% overlap. Increasing this threshold yields more fragmentation in the detection of oscillations, while decreasing this threshold may reduce the accuracy in determining the onset and offset of neural oscillations.”

(4) As with any signal processing method, hyperparameters and their ability to be tuned by the user need to be clearly acknowledged, as they impact the robustness and reproducibility of the method. Here, some of the hyperparameters appear to be: (a) number of cycles around which to construct bounding boxes and (b) overlap percentage of bounding boxes for grouping. Any others should be highlighted by the authors and clearly explained during the course of tool dissemination to the community, ideally in tutorial format through the Github repository.

Response: We thank the reviewer for this helpful suggestion. In response, we added a new subsection that describes the hyper-parameters of CHO as follows:

A new subsection named “Tradeoffs in adjusting the hyper-parameters that govern the detection in CHO”.

“The ability of CHO to detect neural oscillations and determine their fundamental frequency is governed by four principal hyper-parameters. Adjusting these parameters requires understanding their effect on the sensitivity and specificity in the detection of neural oscillations.

The first hyper-parameter is the number of time windows (N in Line 5 in Algorithm 1), that is used to estimate the 1/f noise. In our performance assessment of CHO, we used four windows, resulting in estimation periods of 250 ms in duration for each 1/f spectrum. A higher number of time windows results in smaller estimation periods and thus minimizes the likelihood of observing multiple neural oscillations within this time window, which otherwise could confound the 1/f estimation. However, a higher number of time windows and, thus, smaller time estimation periods may lead to unstable 1/f estimates.

The second hyper-parameter defines the minimum number of cycles of a neural oscillation to be detected by CHO (see Line 23 in Algorithm 1). In our study, we specified this parameter to be two cycles. Increasing the number of cycles increases specificity, as it will reject spurious oscillations. However, increasing the number also reduces sensitivity as it will reject short oscillations.

The third hyper-parameter is the significance threshold that selects positive peaks within the auto-correlation of the signal. The magnitude of the peaks in the auto-correlation indicates the periodicity of the oscillations (see Line 26 in Algorithm 1). Referred to as "NumSTD," this parameter denotes the number of standard errors that a positive peak has to exceed to be selected to be a true oscillation. For this study, we set the "NumSTD" value to 1. Increasing the "NumSTD" value increases specificity in the detection as it reduces the detection of spurious peaks in the auto-correlation. However, increasing the "NumSTD" value also decreases the sensitivity in the detection of neural oscillations with varying instantaneous oscillatory frequencies.

The fourth hyper-parameter is the percentage of overlap between two bounding boxes that trigger their merger (see Line 31 in Algorithm 1). In our study, we set this parameter to 75% overlap. Increasing this threshold yields more fragmentation in the detection of oscillations, while decreasing this threshold may reduce the accuracy in determining the onset and offset of neural oscillations.”

(5) Most of the validation demonstrations in this paper depict the detection capabilities of CHO. For example, the authors demonstrate how to use this tool to reduce false detection of oscillations made up of harmonic activity and show in simulated examples how CHO performs compared to other methods in detection specificity, sensitivity, and accuracy. However, the detection problem is not the same as the 'identity' problem that the paper originally introduced CHO to solve. That is, detecting a non-sinusoidal oscillation well does not help define or characterize its non-sinusoidal 'fingerprint'. An example problem to set up this question is: if there are multiple oscillations at the same base frequency in a dataset, how can their differing harmonic structure be used to distinguish them from each other? To address this at a minimum, Figure 4 (or a followup to it) should simulate signals at similar levels of detectability with different 'identities' (i.e. different levels and/or manifestations of harmonic structure), and evaluate CHO's potential ability to distinguish or cluster them from each other. Then, does a real-world dataset or neuroscientific problem exist in which a similar sort of exercise can be conducted and validated in some way? If the "what" question is to be sufficiently addressed by this tool, then this type of task should be within the scope of its capabilities, and validation within this scenario should be demonstrated in the paper. This is the most fundamental limitation at the paper's current state.

Response: Thank you for your insightful suggestion; we truly appreciate it. We recognize that the 'identity' problem requires further studies to develop appropriate methods. Our current approach does not fully address this issue, as it may detect asymmetric non-sinusoidal oscillations with multiple harmonic peaks, without accounting for different shapes of nonsinusoidal oscillations.

The main reason we could not fully address the “identity” problem results from the general absence of a defined ground truth, i.e., data for which we know the harmonic structure. To overcome this barrier, we would need datasets from well-characterized cognitive tasks or neural disorders. For example, Cole et al. 2017 showed that the harmonic structure of beta oscillations can explain the degree of Parkinson’s disease, and Hu et al. 2023 showed that the number of harmonic peaks can localize the seizure onset zone. Future studies could use the data from these two studies to study whether CHO can distinguish different harmonic structures of pathological neural oscillations.

In this paper, we showed the basic identity of neural oscillations, encompassing elements such as the fundamental frequency and onset/offset. Your valuable insights contribute significantly to our ongoing efforts, and we appreciate your thoughtful consideration of these aspects. In response, we added a new paragraph in the Limitation of the discussion section as below:

“Another limitation of this study is that it does not assess the harmonic structure of neural oscillations. Thus, CHO cannot distinguish between oscillations that have the same fundamental frequency but differ in their non-sinusoidal properties. This limitation stems from the objective of this study, which is to identify the fundamental frequency of non-sinusoidal neural oscillations. Overcoming this limitation requires further studies to improve CHO to distinguish between different non-sinusoidal properties of pathological neural oscillations. The data that is necessary for these further studies could be obtained from the wide range of studies that have linked the harmonic structures in the neural oscillations to various cognitive functions (van Dijk et al., 2010; Schalk, 2015; Mazaheri and Jensen, 2008) and neural disorders (Cole et al., 2017; Jackson et al., 2019; Hu et al., 2023). For example, Cole et al. 2017 showed that a harmonic structure of beta oscillations can explain the degree of Parkinson’s disease, and Hu et al. 2023 showed the number of harmonic peaks can localize the seizure onset zone. “

References:

Beck AM, He M, Gutierrez R, Purdon PL. An iterative search algorithm to identify oscillatory dynamics in neurophysiological time series. bioRxiv. 2022. p. 2022.10.30.514422.

doi:10.1101/2022.10.30.514422

Brady B, Bardouille T. Periodic/Aperiodic parameterization of transient oscillations (PAPTO)Implications for healthy ageing. Neuroimage. 2022;251: 118974.

Fabus MS, Woolrich MW, Warnaby CW, Quinn AJ. Understanding Harmonic Structures Through Instantaneous Frequency. IEEE Open J Signal Process. 2022;3: 320-334.

Jones SR, Pritchett DL, Sikora MA, Stufflebeam SM, Hämäläinen M, Moore CI. Quantitative analysis and biophysically realistic neural modeling of the MEG mu rhythm: rhythmogenesis and modulation of sensory-evoked responses. J Neurophysiol. 2009;102: 3554-3572.

He M, Das P, Hotan G, Purdon PL. Switching state-space modeling of neural signal dynamics. PLoS Comput Biol. 2023;19: e1011395.

Matsuda T, Komaki F. Time Series Decomposition into Oscillation Components and Phase Estimation. Neural Comput. 2017;29: 332-367.

Quinn AJ, Lopes-Dos-Santos V, Huang N, Liang W-K, Juan C-H, Yeh J-R, et al. Within-cycle instantaneous frequency profiles report oscillatory waveform dynamics. J Neurophysiol. 2021;126: 1190-1208.

**Reviewer #2 (Public Review):**
Summary:A new toolbox is presented that builds on previous toolboxes to distinguish between real and spurious oscillatory activity, which can be induced by non-sinusoidal waveshapes. Whilst there are many toolboxes that help to distinguish between 1/f noise and oscillations, not many tools are available that help to distinguish true oscillatory activity from spurious oscillatory activity induced in harmonics of the fundamental frequency by non-sinusoidal waveshapes. The authors present a new algorithm which is based on autocorrelation to separate real from spurious oscillatory activity. The algorithm is extensively validated using synthetic (simulated) data, and various empirical datasets from EEG, intracranial EEG in various locations and domains (i.e. auditory cortex, hippocampus, etc.).Strengths:Distinguishing real from spurious oscillatory activity due to non-sinusoidal waveshapes is an issue that has plagued the field for quite a long time. The presented toolbox addresses this fundamental problem which will be of great use for the community. The paper is written in a very accessible and clear way so that readers less familiar with the intricacies of Fourier transform and signal processing will also be able to follow it. A particular strength is the broad validation of the toolbox, using synthetic, scalp EEG, EcoG, and stereotactic EEG in various locations and paradigms.Weaknesses:At many parts in the results section critical statistical comparisons are missing (e.g. FOOOF vs CHO). Another weakness concerns the methods part which only superficially describes the algorithm. Finally, a weakness is that the algorithm seems to be quite conservative in identifying oscillatory activity which may render it only useful for analysing very strong oscillatory signals i.e.alpha, but less suitable for weaker oscillatory signals (i.e. gamma).

Response: We thank Reviewer #2 for the assistance in improving this manuscript. In the revised manuscript, we have added the missing statistical comparisons, detailed pseudo code, and a subsection that explains the hyper-parameters of CHO. We also recognize the limitations of CHO in detecting gamma oscillations. While our results demonstrate beta-band oscillations in ECoG and EEG signals (see Figures 5 and 6), we had no expectation to find gamma-band oscillations during a simple reaction time task. This is because of the general absence of ECoG electrodes over the occipital cortex, where such gamma-band oscillations may be found.

Nevertheless, our CHO method should be able to detect gamma-band oscillations. This is because if there are gamma-band oscillations, they will be reflected as a bump over the 1/f fit in the power spectrum, and CHO will detect them. We apologize for not specifying the frequency range of the synthetic non-sinusoidal oscillations. The gamma band was also included in our simulation. We added the frequency range (1-40 Hz) of the synthetic nonsinusoidal oscillations in the subsection, the caption of Figure 4, and the result section.

**Reviewer #1 (Recommendations For The Authors):**
(1) The example of a sinusoidal neural oscillation in Fig 1 seems to still exhibit a great deal of nonsinusoidal behavior. Although it is largely symmetrical, it has significant peak-trough symmetry as well as sharper peak structure than typical sinusoidal activity. Nevertheless, it has less harmonic structure than the example on the left. A more precisely-stated claim might be that non-sinusoidal behavior is not the distinguishing characteristic between the two, but rather the degree of harmonic structure.

Response: We are grateful for this thoughtful observation. In response, we now recognize that the depicted example showcases pronounced peak-trough symmetry and sharpness, characteristics that might not be typically associated with sinusoidal behavior. We now better understand that the key differentiator between the examples lies not only in their nonsinusoidal behavior but also in their harmonic structure. To reflect this better understanding, we have refined our manuscript to more accurately articulate the differences in harmonic structure, in accordance with your suggestion. Specifically, we revised the caption of Fig 1 in the manuscript as follows:

The caption of the Fig 1G-L.

“We applied the same statistical test to a more sinusoidal neural oscillation (G). Since this neural oscillation more closely resembles a sinusoidal shape, it does not exhibit any prominent harmonic peaks in the alpha and beta bands within the power spectrum (H) and time-frequency domain (I). Consequently, our test found that the phase of the theta-band and beta-band oscillations were not phase-locked (J-L). Thus, this statistical test suggests the absence of a harmonic structure.”

(2) The statement "This suggests that most of the beta oscillations

detected by conventional methods are simply harmonics of the predominant asymmetric alpha oscillation." is potentially overstated. It is important to constrain this statement to the auditory cortex in which the authors conduct the validation, because true beta still exists elsewhere. The same goes for the beta-gamma claim later on. In general, use of "may be" is also more advisable than the definitive "are".

Response: We thank the reviewer for this thoughtful feedback. To avoid the potential overstatement of our findings we revised our statement on beta oscillations in the manuscript as follows:

Discussion:

“This suggests that most of the beta oscillations detected by conventional methods within auditory cortex may be simply harmonics of the predominant asymmetric alpha oscillation.”

**Reviewer #2 (Recommendations For The Authors):**
All my concerns are medium to minor and I list them as they appear in the manuscript. I do not suggest new experiments or a change in the results, instead I focus on writing issues only.a) Line 50: A reference to the seminal paper by Klimesch et al (2007) on alpha oscillations and inhibition would seem appropriate here.

Response: We added the reference to Klimesch et al. (2007).

b) Figure 4: It is unclear which length for the simulated oscillations was used to generate the data in panels B-G.

Response: We generated oscillations that were 2.5 cycles in length and 1-3 seconds in duration. We added this information to the manuscript as follows.

Figure 4:

“We evaluated CHO by verifying its specificity, sensitivity, and accuracy in detecting the fundamental frequency of non-sinusoidal oscillatory bursts (2.5 cycles, 1–3 seconds long) convolved with 1/f noise.”

Results (page 5, lines 163-165):

“To determine the specificity and sensitivity of CHO in detecting neural oscillations, we applied CHO to synthetic non-sinusoidal oscillatory bursts (2.5 cycles, 1–3 seconds long) convolved with 1/f noise, also known as pink noise, which has a power spectral density that is inversely proportional to the frequency of the signal.”

Methods (page 20, lines 623-626):

“While empirical physiological signals are most appropriate for validating our method, they generally lack the necessary ground truth to characterize neural oscillation with sinusoidal or non-sinusoidal properties. To overcome this limitation, we first validated CHO on synthetic nonsinusoidal oscillatory bursts (2.5 cycles, 1–3 seconds long) convolved with 1/f noise to test the performance of the proposed method.”

c) Figure 5 - supplements: Would be good to re-organize the arrangement of the plots on these figures to facilitate the comparison between Foof and CHO (i.e. by presenting for each participant FOOOF and CHO together).

Response: We combined Figure 5-supplementary figures 1 and 2 into Figure 5-supplementary figure 1, Figure 6-supplementary figures 1 and 2 into Figure 6-supplementary figure 1, and Figure 8-supplementary figures 1 and 2 into Figure 8-supplementary figure 1.

**Author response image 2. sa3fig2:** Figure 5-supplementary figure 1.

**Author response image 3. sa3fig3:** Figure 5-supplementary figure 1.

**Author response image 4. sa3fig4:** Figure 8-supplementary figure 1.

d) Statistics: Almost throughout the results section where the empirical results are described statistical comparisons are missing. For instance, in lines 212-213 the statement that CHO did not detect low gamma while FOOOF did is not backed up by the appropriate statistics. This issue is also evident in all of the following sections (i.e. EEG results, On-offsets of oscillations, SEEG results, Frequency and duration of oscillations). I feel this is probably the most important point that needs to be addressed.

Response: We added statistical comparisons to Figure 5 (ECoG), 6 (EEG), and the results section as follows.

**Author response image 5. sa3fig5:** Validation of CHO in detecting oscillations in ECoG signals. A. We applied CHO and FOOOF to determine the fundamental frequency of oscillations from ECoG signals recorded during the pre-stimulus period of an auditory reaction time task. FOOOF detected oscillations primarily in the alpha- and beta-band over STG and pre-motor area. In contrast, CHO also detected alpha-band oscillations primarily within STG, and more focal beta-band oscillations over the pre-motor area, but not STG. B. We investigated the occurrence of each oscillation within defined cerebral regions across eight ECoG subjects. The horizontal bars and horizontal lines represent the median and median absolute deviation (MAD) of oscillations occurring across the eight subjects. An asterisk (*) indicates statistically significant differences in oscillation detection between CHO and FOOOF (Wilcoxon rank-sum test, p<0.05 after Bonferroni correction).”

**Author response image 6. sa3fig6:** Validation of CHO in detecting oscillations in EEG signals. A. We applied CHO and FOOOF to determine the fundamental frequency of oscillations from EEG signals recorded during the pre-stimulus period of an auditory reaction time task. FOOOF primarily detected alpha-band oscillations over frontal/visual areas and beta-band oscillations across all areas (with a focus on central areas). In contrast, CHO detected alpha-band oscillations primarily within visual areas and detected more focal beta-band oscillations over the pre-motor area, similar to the ECoG results shown in Figure 5. B. We investigated the occurrence of each oscillation within the EEG signals across seven subjects. An asterisk (*) indicates statistically significant differences in oscillation detection between CHO and FOOOF (Wilcoxon rank-sum test, p<0.05 after Bonferroni correction). CHO exhibited lower entropy values of alpha and beta occurrence than FOOOF across 64 channels. C. We compared the performance of FOOO and CHO in detecting oscillation across visual and pre-motor-related EEG channels. CHO detected more alpha and beta oscillations in visual cortex than in pre-motor cortex. FOOOF detected alpha and beta oscillations in visual cortex than in pre-motor cortex.

We added additional explanations of our statistical results to the “Electrocorticographic (ECoG) results” and “Electroencephalographic (EEG) results” sections.

“We compared neural oscillation detection rates between CHO and FOOOF across eight ECoG subjects. We used FreeSurfer to determine the associated cerebral region for each ECoG location. Each subject performed approximately 400 trials of a simple auditory reaction-time task. We analyzed the neural oscillations during the 1.5-second-long pre-stimulus period within each trial. CHO and FOOOF demonstrated statistically comparable results in the theta and alpha bands despite CHO exhibiting smaller median occurrence rates than FOOOF across eight subjects. Notably, within the beta band, excluding specific regions such as precentral, pars opercularis, and caudal middle frontal areas, CHO's beta oscillation detection rate was significantly lower than that of FOOOF (Wilcoxon rank-sum test, p < 0.05 after Bonferroni correction). This suggests comparable detection rates between CHO and FOOOF in premotor and Broca's areas, while the detection of beta oscillations by FOOOF in other regions, such as the temporal area, may represent harmonics of theta or alpha, as illustrated in Figure 5A and B. Furthermore, FOOOF exhibited a higher sensitivity in detecting delta, theta, and low gamma oscillations overall, although both CHO and FOOOF detected only a limited number of oscillations in these frequency bands.”

“We assessed the difference in neural oscillation detection performance between CHO and FOOOF across seven EEG subjects. We used EEG electrode locations according to the 10-10 electrode system and assigned each electrode to the appropriate underlying cortex (e.g., O1 and O2 for the visual cortex). Each subject performed 200 trials of a simple auditory reaction-time task. We analyzed the neural oscillations during the 1.5-second-long pre-stimulus period. In the alpha band, CHO and FOOOF presented statistically comparable outcomes. However, CHO exhibited a greater alpha detection rate for the visual cortex than for the pre-motor cortex, as shown in Figures 6B and C. The entropy of CHO's alpha oscillation occurrences (3.82) was lower than that of FOOOF (4.15), with a maximal entropy across 64 electrodes of 4.16. Furthermore, in the beta band, CHO's entropy (4.05) was smaller than that of FOOOF (4.15). These findings suggest that CHO may offer a more region-specific oscillation detection than FOOOF.

As illustrated in Figure 6C, CHO found fewer alpha oscillations in pre-motor cortex (FC2 and FC4) than in occipital cortex (O1 and O2), while FOOOF found more beta oscillations occurrences in pre-motor cortex (FC2 and FC4) than in occipital cortex. However, FOOOF found more alpha and beta oscillations in visual cortex than in pre-motor cortex.

Consistent with ECoG results, FOOOF demonstrated heightened sensitivity in detecting delta, theta, and low gamma oscillations.

Nonetheless, both CHO and FOOOF identified only a limited number of oscillations in delta and theta frequency bands.

Contrary to the ECoG results, FOOOF found more low gamma oscillations in EEG subjects than in ECoG subjects.”

e) Line 248: The authors find an oscillatory signal in the hippocampus with a frequency at around 8 Hz, which they refer to as alpha. However, several researchers (including myself) may label this fast theta, according to the previous work showing the presence of fast and slow theta oscillations in the human hippocampus (https://pubmed.ncbi.nlm.nih.gov/21538660/, https://pubmed.ncbi.nlm.nih.gov/32424312/).

Response: We replaced “alpha” with “fast theta” in the figure and text. We added a citation for Lega et al. 2012.

f) Line 332: It could also be possible that the auditory alpha rhythms don’t show up in the EEG because a referencing method was used that was not ideal for picking it up. In general, re-referencing is an important preprocessing step that can make the EEG be more susceptible to deep or superficial sources and that should be taken into account when interpreting the data.

Response: We re-referenced our signals using a common median reference (see Methods section). After close inspection of our results, we found that the EEG topography shown in Figure 6 did not show the auditory alpha oscillation because the alpha power of visual locations greatly exceeded that of those locations that reflect oscillations in the auditory cortex. Further, while our statistical analysis shows that CHO detected auditory alpha oscillations, this analysis also shows that CHO detected significantly more visual alpha oscillations.

g) Line 463: It seems that the major limitation of the algorithm lies in its low sensitivity which is discussed by the authors. The authors seem to downplay this a bit by saying that the algorithm works just fine at SNRs that are comparable to alpha oscillations. However, alpha is the strongest single in human EEG which may make the algorithm less suitable for picking up less prominent oscillatory signals, i.e. gamma, theta, ripples, etc. Is CHO only seeing the ‘tip of the iceberg’?

Response: We performed the suggested analysis. For the theta band, this analysis generated convincing statistical results for ECoG signals (Figures 5, 6, and the results section). For theta oscillation detection, we found no statistical difference between CHO and FOOOF. Since FOOOF has a high sensitivity even under SNRs (as shown in our simulation), our analysis suggests that CHO and FOOOF should perform equally well in the detection of theta oscillation, even when the theta oscillation amplitude is small.

To validate the ability of CHO to detect oscillations in high-frequency bands (> 40Hz), such as gamma oscillations and ripples, our follow-up study is applying CHO in the detection of highfrequency oscillations (HFOs) in electrocorticographic signals recorded during seizures. To this end, our follow-up study analyzed 26 seizures from six patients. In this analysis, CHO showed similar sensitivity and specificity as the epileptogenicity index (EI), which is the most commonly used method to detect seizure onset times and zones. The results of this follow-up study were presented at the American Epilepsy Society Meeting in December of 2023, and we are currently preparing a manuscript for submission to a peer-reviewed journal.

In this study, we want to investigate the performance of CHO in detecting the most prominent neural oscillations (e.g., alpha and beta). Future studies will investigate the performance of CHO in detecting more difficult to observe oscillations delta in sleep stages, theta in the hippocampus during memory tasks, and high-frequency oscillation or ripples in seizure or interictal data.

h) Methods: The methods section, especially the one describing the CHO algorithm, is lacking a lot of detail that one usually would like to see in order to rebuild the algorithm themselves. I appreciate that the code is available freely, but that does not, in my opinion, relief the authors of their duty to describe in detail how the algorithm works. This should be fixed before publishing.

Response: We now present pseudo code to describe the algorithms within the new subsection on the hyper-parameterization of CHO.

See Author response image 1.

A new subsection titled “Tradeoffs in adjusting the hyper-parameters that govern the detection in CHO.”

“The ability of CHO to detect neural oscillations and determine their fundamental frequency is governed by four principal hyper-parameters. Adjusting these parameters requires understanding their effect on the sensitivity and specificity in the detection of neural oscillations.

The first hyper-parameter is the number of time windows (N in Line 5 in Algorithm 1), that is used to estimate the 1/f noise. In our performance assessment of CHO, we used four time windows, resulting in estimation periods of 250 ms in duration for each 1/f spectrum. A higher number of time windows results in smaller estimation periods and thus minimizes the likelihood of observing multiple neural oscillations within this time window, which otherwise could confound the 1/f estimation. However, a higher number of time windows and, thus, smaller time estimation periods may lead to unstable 1/f estimates.

The second hyper-parameter defines the minimum number of cycles of a neural oscillation to be detected by CHO (see Line 23 in Algorithm 1). In our study, we specified this parameter to be two cycles. Increasing the number of cycles increases specificity, as it will reject spurious oscillations. However, increasing the number also sensitivity as it will reject short oscillations.

The third hyper-parameter is the significance threshold that selects positive peaks within the auto-correlation of the signal. The magnitude of the peaks in the auto-correlation indicates the periodicity of the oscillations (see Line 26 in Algorithm 1). Referred to as "NumSTD," this parameter denotes the number of standard errors that a positive peak has to exceed to be selected to be a true oscillation. For this study, we set the "NumSTD" value to 1 (the approximate 68% confidence bounds). Increasing the "NumSTD" value increases specificity in the detection as it reduces the detection of spurious peaks in the auto-correlation. However, increasing the "NumSTD" value also decreases the sensitivity in the detection of neural oscillations with varying instantaneous oscillatory frequencies.

The fourth hyper-parameter is the percentage of overlap between two bounding boxes that trigger their merger (see Line 31 in Algorithm 1). In our study, we set this parameter to 75% overlap. Increasing this threshold yields more fragmentation in the detection of oscillations, while decreasing this threshold may reduce the accuracy in determining the onset and offset of neural oscillations.”